

# $\mathcal{N} = 2$ minimal models:
# A holographic needle in a symmetric orbifold haystack

**Alexandre Belin[1] [*], Nathan Benjamin[2] [†], Alejandra Castro[3] [‡],**
**Sarah M. Harrison[4] [§] and Christoph A. Keller[5] [¶]**

**1** CERN, Theory Division, 1 Esplanade des Particules, Geneva 23, CH-1211, Switzerland
**2** Princeton Center for Theoretical Science, Princeton University, Princeton, NJ 08544, USA
**3** Institute for Theoretical Physics, University of Amsterdam, Science Park 904,
Postbus 94485, 1090 GL Amsterdam, The Netherlands
**4** Department of Mathematics and Statistics and Department of Physics,
McGill University, Montreal, QC, Canada
**5** Department of Mathematics, University of Arizona, Tuscon, AZ 85721-0089, USA

[*] a.belin@cern.ch, [†] nathanb@princeton.edu, [‡] a.castro@uva.nl,
[§] sarah.harrison@mcgill.ca, [¶] cakeller@math.arizona.edu

## Abstract

We explore large-$N$ symmetric orbifolds of the $\mathcal{N} = 2$ minimal models, and find evidence that their moduli spaces each contain a supergravity point. We identify single-trace exactly marginal operators that deform them away from the symmetric orbifold locus. We also show that their elliptic genera exhibit slow growth consistent with supergravity spectra in AdS$_3$. We thus propose an infinite family of new holographic CFTs.

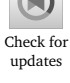
# 1  Introduction

The AdS/CFT correspondence [1–3] is a duality between a conformal field theory in $d$ spacetime dimensions and a theory of quantum gravity that contains Anti-de Sitter space with $d + 1$ spacetime dimensions. One powerful aspect of this correspondence is that it makes tractable how a geometrical description of AdS gravity emerges from the CFT. Still, the fundamental mechanism that would allow us to describe this procedure from first principles is incomplete and unclear. Our aim here is to propose an infinite family of CFTs that have the potential to address aspects of this mechanism.

On the gravitational side of the correspondence, the AdS radius $\ell_{\mathrm{AdS}}$ expressed in Planck units controls the strength of gravitational interactions. To investigate weakly coupled theories of gravity, we thus want this ratio to be large. We are often interested in bulk theories that have the additional property that their low-energy description is given by a local effective field theory (EFT). This EFT could be Einstein gravity or some supergravity variation of it, possibly together with a finite number of matter fields. We then want to know up to what scale $\Lambda$ this EFT description of the bulk is valid. In particular, in order for it to be a useful description, we want it to remain valid far beyond the AdS scale: that is, we want $\Lambda$ to be parametrically larger than $\Lambda_{\mathrm{AdS}}$. The range of validity is closely related to locality of the bulk theory: the EFT description will certainly break down once we reach a scale where the bulk theory ceases to be local. In [4] this was described as sharp holography versus coarse holography.

In the bulk, a typical scenario for such a separation of scales is a string theory setup where the string scale $\Lambda_s$ is parametrically larger than $\Lambda_{\mathrm{AdS}}$. In that case, a supergravity description will remain valid up to $\Lambda_s$. Another scenario is that $\Lambda$ is pushed all the way up to the Planck scale $\Lambda_{\mathrm{Pl}}$, at which point non-perturbative objects such as black holes will certainly spoil the EFT description. Such a scenario could arise for example in a M-theory compactification on an AdS background, or in a putative theory of pure gravity on $\mathrm{AdS}_3$ [5]. On the CFT side, the simplest way to obtain this parametric separation is to demand a large gap for operators with spin greater than two [4,6–9]. On a practical level, if we can construct such a CFT, then it may be possible to describe its holographic dual using supergravity. If on the other hand the CFT does not lead to a separation of scales between $\Lambda$ and $\Lambda_{\mathrm{AdS}}$, then the bulk theory is intrinsically non-local at the AdS scale and would have to be described, for example, by strings.

In this paper, we investigate this question for the $\mathrm{AdS}_3/\mathrm{CFT}_2$ correspondence. Here many of the above statements can be made more precise. For example, the AdS radius is related to

the central charge of the CFT by [10]

$$c = \frac{3\ell_{\text{AdS}}}{2G_N}. \tag{1.1}$$

Semi-classical theories of gravity must therefore be dual to CFTs with a large central charge. To probe for the appearance of a string scale $\Lambda_s$, we can investigate the growth of the CFT spectrum for the low-lying operators. Let us denote by $\rho(\Delta)$ the number of operators at scaling dimension $\Delta$, which is the energy measured in AdS units, $\Delta \sim E/\Lambda_{\text{AdS}}$. If $\Lambda_s \sim \Lambda_{\text{AdS}}$, then for any $E \gg \Lambda_{\text{AdS}}$ we expect the number of states to grow like

$$\textbf{Hagedorn (fast) growth}: \qquad \rho(\Delta) \sim e^{c_H \Delta}. \tag{1.2}$$

In such a case we might even identify the string scale in AdS units with $c_H^{-1}$. In general, a Hagedorn growth of the spectrum indicates that the gravity dual is a string theory in AdS, where the string scale and AdS scale are of the same order. Such a theory would produce large deviations from general relativity at low energies.

On the other hand, if $\Lambda_s \gg \Lambda_{\text{AdS}}$, we can pick $\Lambda_s > E \gg \Lambda_{\text{AdS}}$, and no stringy states contribute such that all states can be described by the EFT. We therefore expect

$$\textbf{Supergravity-like (slow) growth}: \qquad \rho(\Delta) \sim e^{c_S \Delta^\gamma}, \quad \gamma < 1. \tag{1.3}$$

Such supergravity-like theories have many nice properties: in particular they satisfy the sparseness bound which automatically guarantees the familiar thermodynamics of black holes [11], and they have a good chance of producing a bulk EFT that is well approximated by Einstein gravity [12]. Supergravity-like growth is precisely the expected behavior of an Einstein gravity dual. Indeed, supergravity on a background of the type $\text{AdS}_3 \times M_{D-3}$, where the size of $M_{D-3}$ is of the order of the AdS scale, would produce a growth of this type with $\gamma = \frac{D-1}{D}$.

While constructing CFTs that lead to a Hagedorn growth is relatively easy to achieve, it is very difficult to construct CFTs with a supergravity-like growth, and currently only a handful of such CFTs are known. In other words, examples of coarse holography are relatively common, whereas examples of sharp holography are rare. Finding more such sharp needles in the haystack of CFTs is the main goal of this paper.

The specific haystack that we consider in this paper consists of symmetric product orbifolds CFTs. This broad family of CFTs exhibits several of the desired properties of holographic CFTs. The symmetric product orbifold of a seed CFT $X$ is given by

$$\text{Sym}^N(X) := X^N/S_N. \tag{1.4}$$

In the large $N$ limit, these theories have a large central charge and have Hagedorn growth as in (1.2) with $c_H = 2\pi$, regardless of the choice of seed theory $X$ [13]. At first sight it would thus appear that symmetric orbifolds always fix the string scale to the AdS scale, giving intrinsically stringy dualities. This may sound surprising since these theories have a long history of being studied with supergravity tools in the context of AdS/CFT, going back to the study of the D1D5 system [14] compactified on K3 or $T^4$.

A separation of scales can happen because these string theory constructions can have interesting moduli spaces. The symmetric orbifold describes a point where the string length is indeed of the order of the AdS length, so that the gravitational dual is given by tensionless strings [15–19]. To achieve a separation of scales as described above, we need to move far away from that point by deforming the theory: There then exists a strongly coupled regime where the string scale is parametrically large, so that the gravity dual becomes a supergravity theory, such as type IIB supergravity on $\text{AdS}_3 \times S^3$ in the case of the D1D5 system. One can continuously interpolate between these two regimes by tuning an exactly marginal operator

which controls the string length. This is very similar to $AdS_5/CFT_4$, where this coupling is the 't Hooft coupling in $\mathcal{N} = 4$ SYM (and the symmetric group $S_N$ plays the role of the $SU(N)$ gauge group).

To check if for a given $X$ such a separation of scales is possible, we will use two diagnostics. First, we can check if deformations away from the symmetric orbifold point $X^N/S_N$ are even possible in the first place. That is, we can check if the CFT has moduli, i.e. exactly marginal operators that preserve supersymmetry. In fact, to obtain a supergravity growth, we need these moduli to be in the twisted sector since the coupling must introduce interactions between the $N$ copies of the seed. We will argue that in fact the moduli have to be single trace fields, since otherwise their contribution will be suppressed in the large $N$ limit.

Second, we can investigate the growth of states that are protected under deformations. Counting such states (or computing their index) gives a lower bound on the growth of all states. If they themselves exhibit a Hagedorn growth, then there cannot be a supergravity point anywhere in the moduli space [20]. The way to do this in practice is to compute the elliptic genus at the orbifold point,

$$Z_{\text{EG}}(\tau, z) = \text{Tr}_{\text{RR}}\left((-1)^F q^{L_0 - \frac{c}{24}} y^{J_0} \bar{q}^{\bar{L}_0 - \frac{c}{24}}\right), \tag{1.5}$$

which is an index of 1/4-BPS states, and is therefore constant on the entire moduli space.

It is worth pointing out that in our haystack of symmetric orbifold CFTs, most seed CFTs $X$ fail both diagnostics: On the one hand, the weight of states in the twisted sectors, which could potentially serve as deformations, grows linearly in the central charge. This means that if $X$ has central charge $c > 6$, then there cannot be any twisted moduli. On the other hand, [21] showed that if we start with the elliptic genus of almost any seed CFT $X$, then the elliptic genus of its symmetric orbifold will exhibit Hagedorn growth. That is, even for the elliptic genus, a Hagedorn growth is very generic. It is therefore quite difficult to find a sharp needle in the haystack of symmetric orbifold CFTs.

To find a holographic needle, we thus need to circumvent these two hurdles. To deal with the first hurdle, we simply concentrate on CFTs with $c \leq 6$. To deal with the second hurdle, we use the fact that [22–24] found mathematical exceptions to the generic behavior established in [21]: that is, there are specific so-called weak Jacobi forms, which could describe the elliptic genus of a CFT, whose symmetric orbifold exhibits a slow, supergravity-like growth. In this paper we build on these observations to find explicit CFTs that pass both diagnostics: $\mathcal{N} = 2$ minimal models.[1]

## Summary of results

The two main results of this paper are the following:

1. We show that the growth in the elliptic genus for the symmetric orbifold of *any* $\mathcal{N} = 2$ minimal model is supergravity-like, with $\gamma = \frac{1}{2}$.

2. We show that each of these theories has at least one exactly marginal single-trace operator that can turn on a coupling between the copies.

Along the way, we give a complete description of all moduli of these theories. The moduli spaces present interesting structures with several directions, most of which correspond to turning on multi-trace operators. We also discuss the space of slow growing weak Jacobi forms and its relation to the space of physical CFTs. For CFTs with $c < 3$, we conjecture that all slow growing forms that satisfy some basic physical consistency conditions are related to actual

---

[1]The study of minimal models has appeared in other contexts in holography; see for instance [25–30].

CFTs, namely $\mathcal{N} = 2$ minimal models. This strongly suggests that cancellations in the elliptic genus do not arise by mathematical accident. We also comment on a generalization of this conjecture for CFTs with $3 \leq c \leq 6$.

We note that our results are similar in spirit to [31, 32], for which the seed theory has $\mathcal{N} = (2, 2)$ and $c = 6$. In their case, they made a concrete proposal for the supergravity dual. We have not attempted to do so. Since our theories pass the two very restrictive consistency checks outlined above, we believe that there are in fact supergravity duals to them. We are able to write down explicit expression for the BPS spectrum, which will greatly help identifying such gravity duals and could enable a precise matching along the lines of [33, 34].

This paper is organized as follows. In Section 2 we review some basics including the elliptic genus, symmetric orbifolds, slow growth, and exactly marginal operators. In Section 3 we review the $\mathcal{N} = 2$ minimal models, and show that $\mathrm{Sym}^N(\mathcal{N} = 2$ minimal model) has both a slow-growing elliptic genus, and single-trace exactly marginal operators. In Section 4 we generalize to other seed theories, including Kazama-Suzuki theories and tensor products of minimal models. Some detailed calculations are relegated to the appendices.

# 2 Aspects of large $N$ SCFT$_2$

In this section, we review the salient features of $\mathcal{N} = (2, 2)$ two-dimensional CFTs as well as the symmetric orbifolds of such theories in the large $N$ limit. Our discussion here will strengthen and interlace results in prior literature, with the aim to establish necessary conditions that connect symmetric product orbifolds to CFTs that exhibit supergravity-like properties.

## 2.1 $\mathcal{N} = (2, 2)$ SCFTs and the elliptic genus

We will consider unitary compact $\mathcal{N} = (2, 2)$ CFTs in two dimensions, with central charge $c$ and $U(1)_R$ level $\hat{t} = c/6$. Our conventions follow those in, e.g., [35, 36]. The representations of such theories are parametrized by their weight $h$ and their $U(1)_R$ charge $Q$. There are two types of representations: long (or non-BPS) representations, and short (or BPS) representations. A representation is short if it saturates the unitarity bound. In the NS sector, this implies that BPS states are of the form

$$\left| h = \frac{|Q|}{2}, Q \right\rangle_{\mathrm{NS}}, \tag{2.1}$$

and depending on the sign of $Q$, we call it a chiral ($c$) or anti-chiral ($a$) field. In the Ramond sector, this implies that

$$\left| h = \frac{c}{24}, Q \right\rangle_{\mathrm{R}}, \tag{2.2}$$

and we therefore call it a Ramond ground state. Each state has a left- and right-moving component; if both components are BPS, we say the state is 1/2-BPS, and if only one is BPS, we say it is 1/4-BPS.

One of the central objects we will consider is the elliptic genus

$$Z_{\mathrm{EG}}(\tau, z) = \mathrm{Tr}_{\mathrm{RR}} \left( (-1)^F q^{L_0 - \frac{c}{24}} y^{J_0} \bar{q}^{\bar{L}_0 - \frac{c}{24}} \right), \qquad q \equiv e^{2\pi i \tau}, \ \ y \equiv e^{2\pi i z}, \tag{2.3}$$

which captures the 1/4-BPS states in the Ramond sector. If the CFT has a discrete spectrum, then $Z_{\mathrm{EG}}$ is a holomorphic function of $\tau$, as it only receives contributions from right-moving ground states. From this we can define the *NS sector elliptic genus* through

$$Z_{\mathrm{NS}}(\tau, z) = q^{\frac{c}{24}} q^{\frac{\hat{t}}{2}} y^{\hat{t}} Z_{\mathrm{EG}}(\tau, z + \frac{\tau}{2}). \tag{2.4}$$

The shift in $z$ is the half-unit spectral flow that modifies the periodicity of the left-moving fermions from R to NS along the spatial cycle; the right movers stay in the Ramond sector. More generally, the spectral flow automorphism acts on the zero modes as

$$J_0 \to J_0 + 2\hat{t}\,\eta, \qquad L_0 \to L_0 + \eta\,J_0 + \hat{t}\,\eta^2. \tag{2.5}$$

For $\eta \in \mathbb{Z} + 1/2$ it relates the NS (R) sector and the R (NS) sector, while for $\eta \in \mathbb{Z}$ it maps the R and NS sector to themselves. It is this NS sector elliptic genus that will play the central role in counting the growth of states.

Crucially, $Z_{\mathrm{EG}}$ is invariant (up to a phase) under modular transformations. If in addition all $U(1)_R$ charges are integral, then $Z_{\mathrm{EG}}$ is a weak Jacobi form (wJf) of weight 0 and index $\hat{t}$ [37]. The mathematical theory of such wJf was developed in [38]. In this paper, we will consider families of CFTs which may have fractional $U(1)_R$ charges, in which case $Z_{\mathrm{EG}}$ is *not* a wJf. However, there is simple way to convert it to a wJf: If the charges $Q$ of the theory are fractional, this can be done by a procedure we refer to as *unwrapping*, i.e.,

$$Z_{\mathrm{EG}}(\tau, \kappa z) =: \varphi(\tau, z) = \sum_{\substack{n \geq 0 \\ \ell \in \mathbb{Z}}} c(n, \ell) q^n y^\ell, \tag{2.6}$$

where we have simply rescaled $z$ by $\kappa$, which we have chosen as the smallest integer such that $\kappa Q \in \mathbb{Z}$ for all charges $Q$, ensuring charge integrality. Here $\varphi$ is now a wJf of weight 0 whose index $t$ is given in relation to the central charge of the SCFT by

$$t = \frac{c}{6}\kappa^2. \tag{2.7}$$

Assuming the holomorphic $U(1)_R$-symmetry is compact and has rational charges, one can thus always go from an elliptic genus $Z_{\mathrm{EG}}$ of an $\mathcal{N} = (2,2)$ theory with fractional charges to a weak Jacobi form $\varphi$ with an integral Fourier expansion by unwrapping. Note that nothing physical about the theory changed by redefining the $U(1)_R$; in particular, the growth of the spectrum can be analyzed just as well from the unwrapped elliptic genus.

A few additional properties of the wJf will be important and useful in the following sections; for additional background material we refer to [38]. The discriminant of a state $(n, \ell)$ in $\varphi$ is given by $4tn - \ell^2$; states with negative discriminant are called *polar*. Since the weight is 0, specifying all the coefficients $c(n, \ell)$ for the polar states in (2.6) uniquely determines the whole wJf. We will take the most polar state in $\varphi$ to be of the form

$$y^{-b} q^0, \tag{2.8}$$

with $b$ a positive integer and $b \leq t$; we recall that a wJf of index $t$ is allowed to have at most terms with polarity $-t^2$. We will interpret this term, after unwrapping and spectral flow, as the ground state in the NS sector,[2] which leads to the relation

$$t = b\kappa. \tag{2.9}$$

## 2.2 Spectrum of the symmetric orbifold

Beginning with a CFT $X$ of central charge $c$, one can construct an infinite family of CFTs with central charges $cN$, with $N = 1, 2, \ldots$, by taking symmetric product orbifolds of $X$. The $N$-th

---

[2]It is possible that the contribution of the NS vacuum to the elliptic genus vanishes due to fermionic zero modes in the RR sector. An example where this occurs is the non-linear sigma models on an odd-dimensional Calabi-Yau [39]. A more extreme version of these cancellations is the non-linear sigma model with target space $T^4$ where the elliptic genus vanishes alltogether. We will not consider such theories in this paper and always assume that the vacuum gives a non-vanishing contribution to the elliptic genus.

symmetric product of $X$, which we will denote as $\mathrm{Sym}^N(X)$, is constructed by tensoring $N$ copies of the CFT $X$ with each other and then orbifolding by the symmetric group $S_N$, i.e.,

$$\mathrm{Sym}^N(X) = \underbrace{X \otimes \ldots \otimes X}_{N}/S_N. \tag{2.10}$$

One very appealing aspect of this construction is that several quantities can be easily expressed in terms of the "seed theory" $X$. In particular, the partition function (elliptic genus) of the symmetric product theory $\mathrm{Sym}^N(X)$ is completely determined by the partition function (elliptic genus) of the seed theory. This feature is elegantly captured by the generating function for the elliptic genera of $\mathrm{Sym}^N(X)$ [40], which shows

$$\sum_{N=0}^{\infty} p^N Z_{\mathrm{EG}}^N(\tau, \kappa z) = \prod_{\substack{m>0 \\ n,\ell}} \frac{1}{(1 - p^m q^n y^\ell)^{c(mn,\ell)}}, \tag{2.11}$$

where $Z_{\mathrm{EG}}^N$ is the elliptic genus of $\mathrm{Sym}^N(X)$, and the coefficients $c(n,\ell)$ are those in the elliptic genus of the seed $X$. To avoid introducing further notation, we are displaying the generating function for the unwrapped elliptic genus, albeit this product relation also holds without the need of unwrapping. This expression is robust under spectral flow as well, allowing us to obtain the NS sector spectrum via (2.4).

From these generating functions, one can read off the spectrum of $\mathrm{Sym}^N(X)$ in the large $N$ limit. In the case of the partition function, the behavior of the spectrum is universal and has Hagedorn growth, independent of the choice of the seed theory [13]. On the other hand, the elliptic genus may exhibit more interesting behavior. There are generically cancellations among states due to the presence of the $(-1)^F$ term in (2.6), and these cancellations may be so large that the spectrum of 1/4-BPS states contributing to the elliptic genus of $\mathrm{Sym}^N(X)$ significantly deviates from the spectrum of the partition function at large $N$. We distinguish between two cases: we say the elliptic genus of the symmetric product has "fast growth" if the growth of the coefficients is Hagedorn (1.2), and we say it has "slow growth" if the coefficients exhibit supergravity-like behavior (1.3).

The authors of [23, 24] give a simple criterion to determine which seed theories $X$ yield elliptic genera which exhibit slow growth at large $N$. Let us briefly state this criterion, which is stated for the unwrapped elliptic genus (2.6). Let $\varphi(\tau, z)$ be the seed wJf of weight 0 and index $t$. The criterion then works the following way: Assume that $q^0 y^b$ is the most polar term of $\varphi$ for some $b \leq t$. To determine the growth of the symmetric orbifold, for each term $q^n y^\ell$ in the seed wJf with non-vanishing coefficient $c(n,\ell)$, we compute the quantity

$$\alpha = \max_{j=0,\ldots,b-1} \left( -\frac{t}{b^2} j \left( j - \frac{b\ell}{t} \right) - n \right). \tag{2.12}$$

If for any term $\alpha > 0$, the symmetric orbifold of $\varphi$ has Hagedorn growth; otherwise, it has supergravity-like growth. Hence, a necessary condition on our needles is to have $\alpha \leq 0$ for all states. Note that this is a condition only for polar states: states with $4tn - \ell^2 \geq 0$ automatically give $\alpha < 0$. This implies that for a fixed $t$, there is only a relatively small number of such terms, roughly $\sim t^2/12$.

The analysis of (2.12) in [23] also established that $b^2 > t$ implies Hagedorn growth. This implies that if $\varphi$ is the elliptic genus of a bona fide CFT, i.e. $\kappa = 1$ in (2.9), all such forms will have Hagedorn growth except if $t = b = 1$. This is the case of the K3 sigma model, which does exhibit supergravity-like growth in the symmetric product elliptic genus [21]. If $\varphi$ however is an unwrapped elliptic genus, there are wJf that meet the criteria of supergravity-like growth, and one necessary condition for their existence is that

$$c = \frac{6b^2}{t} \leq 6, \tag{2.13}$$

which is already a strong restriction on the central charge of the seed CFT. In the remainder of this section we will analyze other criteria that the needles in the haystack of $\text{Sym}^N(X)$ need to satisfy and how they intertwine with the criteria imposed by (2.12).

## 2.3 Marginal operators and the large $N$ limit

We described above that for some theories, there is a discrepancy between a slow (supergravity-like) growing elliptic genus and a fast (Hagedorn) growing partition function. In this section we argue that this discrepancy is explained by the existence of a moduli space for the theory. That is, we want to establish the existence of suitable marginal operators that allow us to deform the theory.

Our reasoning is based on known string theory constructions. For example, the symmetric product orbifold of a K3 sigma model, $\text{K3}^N/S_N$, describes the D1D5 system on $\text{AdS}_3 \times S^3 \times \text{K3}$ when the string length is of order the AdS length [41,42]. The symmetric product CFT contains a marginal operator which can drive the system into a strongly coupled regime; this separates the string and AdS lengths, and leads to the supergravity description of D1D5. In particular, the marginal deformation reduces the Hagedorn growth by introducing large anomalous dimensions to most of the operators in theory. The elliptic genus however, is protected under this deformation and hence already captures the reductions that match the BPS spectrum of the supergravity theory [33]. This makes the elliptic genus a precursor to quantify the gravitational features; identifying the marginal operator that turns on the coupling further strengthens the argument by giving a physical explanation to the reduced growth in the elliptic genus.

Let us describe in more detail the properties of these marginal operators that introduce the separation of scales at large $N$. First, it is instructive to highlight the resemblance of symmetric orbifolds of 2d CFTs and their large $N$ limit to the well known large $N$ limit of $\mathcal{N} = 4$ supersymmetric Yang-Mills (SYM) in 4d. The symmetric group $S_N$ plays the role of the gauge group $SU(N)$. Similarly, there is a notion of single-trace and multi-trace states:

**Single trace:** a single-trace state is either a symmetrized state of the seed theory (which is in the untwisted sector), or a single cycle of some length $L$ in the twisted sector.

**Multi-trace:** A general (multi-trace) state in the $S_N$ orbifold is then the (symmetrized) product of at least two single-trace factors.

The reason for this terminology is that in the large $N$ limit, their correlation functions behave in the same way as correlation functions of SYM: to leading order in $N$, they are given by combinatorial Wick contractions of all single-trace factors [43, 44].

The generalization of the string length for K3 is the following. Symmetric orbifolds can have a moduli space. That is, they may contain operators $\mathcal{O}$ such that a deformation to the action by

$$\lambda N^{\frac{\beta}{2}} \int d^2x\, \mathcal{O}(x), \tag{2.14}$$

gives another CFT. That is, we can use these $\mathcal{O}$ to move around on the moduli space. We take $\mathcal{O}$ to be normalized such that it has a unit two-point function, but allow for the moment arbitrary powers of $N$ multiplying $\lambda$, which we take to be an $N$-independent coupling. To preserve conformal invariance, $\mathcal{O}$ has to be exactly marginal, i.e. it must have conformal dimension $(h, \bar{h}) = (1, 1)$, and not receive any corrections to the dimension to all orders in perturbation theory. If we want it to preserve $\mathcal{N} = (2, 2)$ supersymmetry, it additionally must be the $G_{-1/2}^-$ ($G_{-1/2}^+$) descendant of a (anti-)chiral primary in the NS sector of

$$Q = 1(-1), \qquad h = 1/2. \tag{2.15}$$

The moduli are thus given by primary operators with (2.15), which may be of the form $(c,c)$, $(a,c)$, $(c,a)$, or $(a,a)$.

A priori, we can choose $\beta$ in (2.14) any way we want. However, we want to ensure that our theory has a good planar limit for $N \to \infty$. This only happens for an appropriate choice of $\beta$. In the case of 4d SYM, this corresponds to the statement that the 't Hooft coupling $\lambda$ is related to the Yang-Mills coupling by $\lambda = Ng^2$. For symmetric orbifolds, the situation is more complicated. As we review in appendix D, the correct choice of $\beta$ depends on the type of multi-trace operator $\mathcal{O}$. In particular, when $\mathcal{O}$ is a $K$-trace operator, we have

$$\beta = 2 - K. \tag{2.16}$$

It then turns out that to leading order in $N$, only single-trace moduli lead to significant deformations of the CFT. All other moduli contribute $O(N^{-1})$ corrections to the spectrum. We will therefore be mostly interested in single-trace moduli.

## 2.4 Twisted sector moduli

Having established how single trace moduli can deform the symmetric orbifold, let us now explain how to count them. To identify such exactly marginal operators, it is useful to work in the Ramond sector. If we have chiral primaries in the NS sector of $\text{Sym}^N(X)$ obeying (2.15), then the corresponding Ramond ground states will have

$$h = \frac{cN}{24}, \qquad Q = 1 - \frac{cN}{6}, \tag{2.17}$$

if the primary is chiral, and

$$h = \frac{cN}{24}, \qquad Q = -1 + \frac{cN}{6}, \tag{2.18}$$

for anti-chiral primaries. These are the operators we want to detect and quantify, which as we will see are straightforward to count.

The Ramond ground states are 1/2-BPS states; for a seed CFT these are captured by the function

$$Z_{\frac{1}{2}-\text{BPS}}(y, \bar{y}) = \sum_{Q, \bar{Q}} d(Q, \bar{Q}) y^Q \bar{y}^{\bar{Q}}, \tag{2.19}$$

where $d(Q, \bar{Q})$ is the multiplicity of the RR ground state of charge $(Q, \bar{Q})$. To compute the spectrum of 1/2-BPS states for the $N$-th symmetric orbifold, we use the generating function (see, e.g. [33, 45]),

$$\sum_{N=0}^{\infty} Z_{\frac{1}{2}-\text{BPS}}^N(y, \bar{y}) p^N = \prod_{L=1}^{\infty} \prod_{Q, \bar{Q}} \frac{1}{(1 - p^L y^Q \bar{y}^{\bar{Q}})^{d(Q,\bar{Q})}}, \tag{2.20}$$

where $Z_{\frac{1}{2}-\text{BPS}}^N(y, \bar{y})$ is the 1/2-BPS spectrum of $\text{Sym}^N(X)$. To identify the twisted sector states, we can use a modified version of (2.20),

$$\sum_{N=0}^{\infty} Z_{\frac{1}{2}-\text{BPS}}^{N,L}(y, \bar{y}) p^N = \prod_{L=1}^{L_{\max}} \prod_{Q, \bar{Q}} \frac{1}{(1 - p^L y^Q \bar{y}^{\bar{Q}})^{d(Q,\bar{Q})}}, \tag{2.21}$$

which only counts states that have no twist cycle longer than $L_{\max}$. In particular, $L_{\max} = 1$ gives the untwisted states. Finally, for a given $N$ we can identify which of these states ground states carry charges as in (2.17)-(2.18). These formulas allows us to read off the number of moduli of a given chirality type from (2.20) for any $N$, and if it is single or multi trace depending

on which term in the product formula leads to that state. Note that this number vanishes for $N = 0$, and can increase with $N$. In fact, for symmetric orbifolds, this number stabilizes above a certain value of $N$, see e.g. [13].

Let us focus on the $(c, c)$ moduli, and explain how to count them. The generating function for $(c, c)$ primaries is

$$\sum_{N=0}^{\infty} Z_{cc}^{N}(y, \bar{y}) p^{N} = \prod_{L=1}^{\infty} \prod_{Q, \bar{Q}} \frac{1}{(1 - p^{L} y^{Q+cL/6} \bar{y}^{\bar{Q}+cL/6})^{d(Q, \bar{Q})}}, \tag{2.22}$$

which is the appropriate spectral flow of (2.20) to the NS sector 1/2-BPS states. The lightest moduli in a given twisted sector would potentially come from the NS ground state of the seed: This corresponds to the values $Q = -c/6$ in (2.22). We find that in the NS sector, the twist $L$ BPS operator with the smallest charge has

$$Q = \frac{c}{6}(L - 1), \tag{2.23}$$

from which it immediately follows that its weight is

$$h = \frac{c}{12}(L - 1). \tag{2.24}$$

Note that for bosonic theories, the well-known expression for the weight of the twist operator is $h = \frac{c}{24}(L - 1/L)$, see e.g. [46, 47]; to impose the correct monodromy for fermions while preserving supersymmetry, it is necessary to include a spin field, whose weight increases the weight of the twist operator. As an immediate consequence of (2.24), we see that the highest value of $c$ for which there can be twisted moduli is $c = 6$, in which case there can be twist-2 moduli. Therefore, the existence of a marginal operator that preserves supersymmetry leads to

$$c \leq 6. \tag{2.25}$$

This agrees nicely, and non-trivially, with what we found in (2.13). We reiterate that the argument here is valid for the $(c, c)$ moduli, and one can proceed in a similar fashion for the other types of moduli.

# 3 The symmetric product of $\mathcal{N} = 2$ minimal models

From Section 2, we have summarized the two necessary conditions we require for a symmetric product theory to have a semiclassical gravity dual – sub-Hagedorn growth in the elliptic genus, and (at least one) single-trace exactly marginal operator – both require the seed theory to have $c \leq 6$. In the rest of this paper, we will analyze various $\mathcal{N} = (2, 2)$ CFTs with $c \leq 6$ as candidate seed theories. Unfortunately $\mathcal{N} = (2, 2)$ CFTs with $3 \leq c \leq 6$ are unclassified, but for $c < 3$ a classification is complete and is given by the minimal models. In this section we will study the symmetric product of $\mathcal{N} = 2$ minimal models. Remarkably we will find that the symmetric product of *every* unitary $\mathcal{N} = 2$ minimal model obeys both of our conditions for a semiclassical gravity dual. In Section 3.1 we review salient features of the minimal models. In Section 3.2 we describe the supergravity-like growth of their symmetric product elliptic genus. In Section 3.3 we describe their exactly marginal operators.

## 3.1 The $\mathcal{N} = 2$ minimal models

The unitary $\mathcal{N} = (2, 2)$ SCFTs with $c < 3$ are fully classified by the $\mathcal{N} = 2$ minimal models [48, 49]. They come labeled by a positive integer $k$, with the central charge given by

$$c = \frac{3k}{k + 2}. \tag{3.1}$$

Such minimal models are rational CFTs; that is, they have a finite number of irreducible representations of the $\mathcal{N} = 2$ superconformal Virasoro algebra. The characters of the algebra at these values of the central charge are combined into modular invariant partition functions. The possible ways of obtaining such partition functions are given by an ADE classification of the theories. Originally, this ADE classification was found for the $A_1^{(1)}$ WZW models [50]: all modular invariant partition functions are given by combinations of their characters $\chi_r(\tau)$,

$$Z^{A_1^{(1)}}(\tau, \bar{\tau}) = \sum_{1 \le r, r' < k+1} N_{r,r'}^{\Phi} \chi_r(\tau) \chi_{r'}(\bar{\tau}). \tag{3.2}$$

Here $\Phi$ is one of the simply laced Dynkin diagrams, which are given by the $A$, $D$, and $E$ series. The allowed multiplicity matrices $N_{r,r'}^{\Phi}$ are in one-to-one correspondence to such Dynkin diagrams; this explains the term 'ADE classification'. The same ADE classification as in the $A_1^{(1)}$ WZW models can be used to obtain modular invariants of $\mathcal{N} = 2$ minimal models [51]. In general there are many more modular invariants [52], but if we require the CFT to be invariant under spectral flow by half a unit, then the possible invariants are indeed classified by the same ADE series as the $A_1^{(1)}$ WZW models [53, 54]. (In string theory, this condition is usual phrased as preserving spacetime supersymmetry.) Since we rely on spectral flow for our arguments, we will restrict ourselves to such invariants. Their partition functions in the Ramond sector are then given by

$$Z_{\text{RR}}^{\Phi}(\tau, \bar{\tau}, z, \bar{z}) = \frac{1}{2} \sum_{1 \le r, r' \le k+1} N_{r,r'}^{\Phi} \sum_{s \in \mathbb{Z}/(2k+2)\mathbb{Z}} \tilde{\chi}_s^r(\tau, z) \tilde{\chi}_s^{r'}(\bar{\tau}, \bar{z}), \tag{3.3}$$

where $\tilde{\chi}_s^r(\tau, z)$ are related to characters of the $\mathcal{N} = 2$ algebra at these values of the central charge. See Appendix A.1 for more details. The upshot of our discussion is that the minimal models come in the following families:

- the $A$-series, which have $c = \frac{3k}{k+2}$ for any positive integer $k$ and are denoted $A_{k+1}$;

- the $D$-series, which have $c = \frac{3k}{k+2}$ for any even $k \ge 4$ and are denoted $D_{k/2+2}$;

- and three exceptional theories denoted $E_6$, $E_7$, and $E_8$, which have $c = \frac{5}{2}, \frac{8}{3}$, and $\frac{14}{5}$ respectively.

From the partition function (3.3), we can easily recover both the elliptic genus and the 1/2-BPS spectrum. To recover the elliptic genus of the $\Phi$-type minimal model we set $\bar{z} = 0$, giving

$$Z_{\text{EG}}^{\Phi}(\tau, z) = \frac{1}{2} \sum_{1 \le r, r' \le k+1} N_{r,r'}^{\Phi} \left( \tilde{\chi}_{r'}^r(\tau, z) - \tilde{\chi}_{-r'}^r(\tau, z) \right). \tag{3.4}$$

To recover the 1/2-BPS partition function $Z_{\frac{1}{2}-\text{BPS}}^{\Phi}$, we specialize $q = \bar{q} = 0$, giving

$$Z_{\frac{1}{2}-\text{BPS}}^{\Phi}(y, \bar{y}) = \frac{1}{2} \sum_{1 \le r \le k+1} N_{r,r}^{\Phi} \left( (y\bar{y})^{\frac{r}{k+2} - \frac{1}{2}} + (y\bar{y})^{-\frac{r}{k+2} + \frac{1}{2}} \right). \tag{3.5}$$

## 3.2 Growth of symmetric product of minimal models

Using (3.4), we can obtain expressions for the elliptic genus of the minimal models. They turn out to be given by [55, 56]:

$$Z_{\text{EG}}^{A_{k+1}}(\tau, z) = \frac{\theta_1\left(\tau, \frac{(k+1)z}{k+2}\right)}{\theta_1\left(\tau, \frac{z}{k+2}\right)}, \qquad A\text{-series,}$$

$$Z_{\text{EG}}^{D_{k/2+2}}(\tau,z) = \frac{\theta_1\left(\tau,\frac{kz}{k+2}\right)\theta_1\left(\tau,\frac{(k+4)z}{2(k+2)}\right)}{\theta_1\left(\tau,\frac{2z}{k+2}\right)\theta_1\left(\tau,\frac{kz}{2(k+2)}\right)}, \qquad D\text{-series,}$$

$$Z_{\text{EG}}^{E_6}(\tau,z) = \frac{\theta_1\left(\tau,\frac{3z}{4}\right)\theta_1\left(\tau,\frac{2z}{3}\right)}{\theta_1\left(\tau,\frac{z}{4}\right)\theta_1\left(\tau,\frac{z}{3}\right)}, \qquad E_6,$$

$$Z_{\text{EG}}^{E_7}(\tau,z) = \frac{\theta_1\left(\tau,\frac{7z}{9}\right)\theta_1\left(\tau,\frac{2z}{3}\right)}{\theta_1\left(\tau,\frac{2z}{9}\right)\theta_1\left(\tau,\frac{z}{3}\right)}, \qquad E_7,$$

$$Z_{\text{EG}}^{E_8}(\tau,z) = \frac{\theta_1\left(\tau,\frac{4z}{5}\right)\theta_1\left(\tau,\frac{2z}{3}\right)}{\theta_1\left(\tau,\frac{z}{5}\right)\theta_1\left(\tau,\frac{z}{3}\right)}, \qquad E_8, \tag{3.6}$$

with the convention that the supercharge has charge $\pm1$. We define $\theta_1(\tau,z)$ as the usual Jacobi theta function:

$$\theta_1(\tau,z) = -iq^{\frac{1}{8}}y^{\frac{1}{2}}\prod_{n=1}^{\infty}(1-q^n)(1-yq^n)(1-y^{-1}q^{n-1}). \tag{3.7}$$

We now want to rescale so that all the charges in $Z_{\text{EG}}(\tau,z)$ are integers with gcd 1, according to (2.6). This gives the following weak Jacobi forms:

$$\varphi^{A_{k+1}}(\tau,z) = \frac{\theta_1(\tau,(k+1)z)}{\theta_1(\tau,z)}, \qquad b = \frac{k}{2}, \;\; t = \frac{k(k+2)}{2}: \quad A\text{-series, } k \text{ even,}$$

$$\varphi^{A_{k+1}}(\tau,z) = \frac{\theta_1(\tau,2(k+1)z)}{\theta_1(\tau,2z)}, \qquad b = k, \;\; t = 2k(k+2): \quad A\text{-series, } k \text{ odd,}$$

$$\varphi^{D_{k/2+2}}(\tau,z) = \frac{\theta_1\left(\tau,\frac{kz}{2}\right)\theta_1\left(\tau,\frac{(k+4)z}{4}\right)}{\theta_1\left(\tau,\frac{kz}{4}\right)\theta_1(\tau,z)}, \;\; b = \frac{k}{4}, \;\; t = \frac{k(k+2)}{8}: \quad D\text{-series, } k \equiv 0 \;(\text{mod } 4),$$

$$\varphi^{D_{k/2+2}}(\tau,z) = \frac{\theta_1(\tau,kz)\theta_1\left(\tau,\frac{(k+4)z}{2}\right)}{\theta_1\left(\tau,\frac{kz}{2}\right)\theta_1(\tau,2z)}, \;\; b = \frac{k}{2}, \;\; t = \frac{k(k+2)}{2}: \quad D\text{-series, } k \equiv 2 \;(\text{mod } 4),$$

$$\varphi^{E_6}(\tau,z) = \frac{\theta_1(\tau,8z)\theta_1(\tau,9z)}{\theta_1(\tau,4z)\theta_1(\tau,3z)}, \qquad b = 5, \;\; t = 60: \qquad E_6,$$

$$\varphi^{E_7}(\tau,z) = \frac{\theta_1(\tau,6z)\theta_1(\tau,7z)}{\theta_1(\tau,2z)\theta_1(\tau,3z)}, \qquad b = 4, \;\; t = 36: \qquad E_7,$$

$$\varphi^{E_8}(\tau,z) = \frac{\theta_1(\tau,12z)\theta_1(\tau,10z)}{\theta_1(\tau,5z)\theta_1(\tau,3z)}, \qquad b = 7, \;\; t = 105: \qquad E_8. \tag{3.8}$$

The parameters $b$ and $t$ are defined in (2.7)-(2.9).

Remarkably, *every* weak Jacobi form in (3.8) satisfies the condition in (2.12) and therefore is a wJf that exhibits slow growth in the symmetric product! In Appendix B we prove this explicitly.[3] Moreover, we can write down a generating function for the low-lying states in the large $N$ symmetric product which exhibit this slow growth. The $q^h y^\ell$ term in the NS-sector elliptic genus of the symmetric product can be formed into a generating function we call $\chi_\infty^{\text{NS}}(q,y)$; see [23]. In Appendix C, we give explicit expressions for $\chi_\infty^{\text{NS}}(q,y)$ for all minimal models. They all take the qualitative form

$$\chi_\infty^{\text{NS}}(q,y) = \prod_{h,\ell}\frac{1}{(1-q^h y^\ell)^{f_{\text{NS}}(h,\ell)}}, \tag{3.9}$$

---

[3]We note that there is also a simpler proof of this that does not rely on (2.12) [57].

where $f_{\mathrm{NS}}(h, \ell)$ takes only a finite number of allowed values. This means that the $q^h$ coefficient of the NS-sector elliptic genus grows roughly as $\sim e^{\sqrt{h}}$, which is indeed supergravity-like slow growth with $\gamma = 1/2$ in (1.3). In contrast, if the theory had Hagedorn growth, the $f_{\mathrm{NS}}(h, \ell)$ would grow exponentially. We emphasize functions $\chi_{\mathrm{NS}}^{\infty}(q, y)$ are *not* counting consistent CFT spectra; instead they are counting the states that have energy much less than $N$ in the large $N$ limit. They are counting low-lying states in the theory, for instance Kaluza-Klein modes in a dimensional reduction of the supergravity theory.

We therefore arrive at one of the main results of this paper: **The symmetric product orbifold of any $\mathcal{N} = 2$ minimal model exhibits sub-Hagedorn growth in the NS-sector elliptic genus.** This is a necessary and very nontrivial condition for the theory to have a large-radius Einstein gravity locus in moduli space.

## 3.3 Moduli

Let us now investigate the existence of moduli in symmetric orbifolds of minimal models along the lines of our discussion in Sections 2.3 and 2.4. Note that similar to the case of K3 or $T^4$, we take 'moduli' to mean fields which preserve both conformal invariance and supersymmetry. (There are exactly marginal fields such as $J\bar{J}$ which do not preserve supersymmetry [58–60].) That is, we want to study what is usually called the moduli space, and not the conformal manifold of the theory.

In the case of K3, the seed theory itself already has 80 moduli (which can be seen from the Hodge diamond of K3). The symmetric orbifold then automatically has symmetrized versions of these in the untwisted sector. In addition, there are 4 more moduli in the twist-2 sector. These are the moduli that actually change the string length. In total there are thus 84 moduli. See, for example, [42].

For minimal models, the situation is different: the seed theory has no supersymmetry-preserving moduli. It does contain relevant chiral fields though, which potentially can be combined to give marginal fields in the symmetric orbifold theory, giving multi-trace moduli in the untwisted sector. Moreover moduli can also appear in the twisted sector. We will see that both in fact happen.

**A-series**

For $\Phi = A_{k+1}$, we find the following expression for the 1/2-BPS partition function:

$$Z_{\frac{1}{2}-\mathrm{BPS}}^{A_{k+1}}(y, \bar{y}) = \sum_{j=1}^{k+1} (y\bar{y})^{\frac{j}{k+2} - \frac{1}{2}}. \tag{3.10}$$

We give detailed expressions for the moduli in appendix A.2. Here let us simply summarize the counting in Table 1, and point out that for every value of $k$ we always find at least one single traced modulus in the twisted sector.

**D-series**

For $\Phi = D_{k/2+2}$, we find

$$Z_{\frac{1}{2}-\mathrm{BPS}}^{D_{k/2+2}}(y, \bar{y}) = 1 + \sum_{j=1}^{\frac{k}{2}+1} (y\bar{y})^{\frac{2j-1}{k+2} - \frac{1}{2}}. \tag{3.11}$$

The analysis of the $D$ series is similar to the $A$ series with $k$ even, and a summary is given in Table 1. Again we note that there is always a single trace moduli in the twist 3 sector.

*E*-series

For the E-type minimal models, we find

$$Z^{E_6}_{\frac{1}{2}-\text{BPS}}(y,\bar{y}) = (y\bar{y})^{-\frac{5}{12}} + (y\bar{y})^{-\frac{1}{6}} + (y\bar{y})^{-\frac{1}{12}} + (y\bar{y})^{\frac{1}{12}} + (y\bar{y})^{\frac{1}{6}} + (y\bar{y})^{\frac{5}{12}},$$

$$Z^{E_7}_{\frac{1}{2}-\text{BPS}}(y,\bar{y}) = (y\bar{y})^{-\frac{4}{9}} + (y\bar{y})^{-\frac{2}{9}} + (y\bar{y})^{-\frac{1}{9}} + 1 + (y\bar{y})^{\frac{1}{9}} + (y\bar{y})^{\frac{2}{9}} + (y\bar{y})^{\frac{4}{9}}, \qquad (3.12)$$

$$Z^{E_8}_{\frac{1}{2}-\text{BPS}}(y,\bar{y}) = (y\bar{y})^{-\frac{7}{15}} + (y\bar{y})^{-\frac{4}{15}} + (y\bar{y})^{-\frac{2}{15}} + (y\bar{y})^{-\frac{1}{15}} +$$

$$+ (y\bar{y})^{\frac{1}{15}} + (y\bar{y})^{\frac{2}{15}} + (y\bar{y})^{\frac{4}{15}} + (y\bar{y})^{\frac{7}{15}}.$$

The number of moduli can be computed directly via (2.20). It turns out that all three *E* series models have one single trace modulus in the twist 2 sector.

Table 1: Number of moduli for symmetric orbifolds of the *ADE* minimal models. We always take $N$ large enough so that the moduli have converged. $P(n)$ is the integers partition function, i.e. $\sum_{n=0}^{\infty} P(n)q^n = \prod_{n=1}^{\infty} \frac{1}{(1-q^n)}$.

| Series | $k$ | untwisted moduli | twisted moduli | single trace twisted |
|---|---|---|---|---|
| $A_2$ | 1 | 1 | 28 | 1 twist 5, 1 twist 7 |
| $A_3$ | 2 | 3 | 26 | 1 twist 3, 1 twist 4, 1 twist 5 |
| $A_5$ | 4 | 9 | 24 | 1 twist 2, 1 twist 3, 1 twist 4 |
| $A_{k+1}$ | odd, $\geq 3$ | $P(k+2)-2$ | 9 | 1 twist 3 |
| $A_{k+1}$ | even, $\geq 6$ | $P(k+2)-2$ | $10 + \sum_{r=1}^{\frac{k}{2}+2} P(r)$ | 1 twist 2, 1 twist 3 |
| $D_4$ | 4 | 6 | 20 | 1 twist 2, 2 twist 3, 1 twist 4 |
| $D_{\frac{k}{2}+2}$ | 0 mod 4, $\geq 8$ | $P(\frac{k}{2}+1) + P(\frac{k}{4}+1)$ | $8 + \sum_{r=1}^{\frac{k}{4}+1} P(r)$ | 1 twist 2, 1 twist 3 |
| $D_{\frac{k}{2}+2}$ | 2 mod 4, $\geq 6$ | $P(\frac{k}{2}+1)$ | 7 | 1 twist 3 |
| $E_6$ | 10 | 4 | 5 | 1 twist 2 |
| $E_7$ | 16 | 6 | 5 | 1 twist 2 |
| $E_8$ | 28 | 6 | 5 | 1 twist 2 |

We summarize our results in Table 1. The upshot is that the symmetric orbifold of all minimal models in the ADE series have one or more single trace moduli in the twisted sector. We can thus deform the theory away from the orbifold point, and potentially reach a supergravity point in the moduli space.

# 4 The landscape of symmetric orbifold theories

## 4.1 A conjecture on the landscape

In the prior sections we established that the elliptic genus of any minimal model can be unwrapped to give a weak Jacobi form of index $t$ with maximal polar term $q^0 y^b$ that is slow growing, where $t$ and $b$ are related by

$$c = \frac{6b^2}{t}. \qquad (4.1)$$

Let us now address the converse of that statement: Does every slow growing form come from the elliptic genus of a $\mathcal{N} = (2,2)$ CFT?

Before we can make a precise statement, let us first discuss several qualifications. First we note that due to (2.9), any unwrapping of an elliptic genus automatically gives

$$\frac{t}{b} \in \mathbb{Z}. \tag{4.2}$$

Any conjecture we are making can thus only hold for wJf that satisfy (4.2).

Next, we note that slow growing wJf form a vector space, whereas CFTs do not. The best we can hope for is thus that elliptic genera of minimal models may give a basis for the space of slow growing forms. More precisely, consider the space of wJf of weight 0 and index $t$, $J_{0,t}$, and for fixed $t$ and $b$ we define

$$U_{t,b} := \{\varphi \in J_{0,t} : \varphi \text{ has no terms more polar than } y^b q^0, \text{ satisfies } \alpha \leq 0 \text{ w.r.t } b\}, \tag{4.3}$$

that is the vector space of wJf that satisfy the slow growth condition in (2.12) with respect to $b$, and that do not have any terms more polar than $y^b q^0$. Note that because we want $U_{t,b}$ to be a vector space, we have to allow for the coefficient of $y^b q^0$ to vanish. This is no problem formally, since we can still check the $\alpha$ condition for such a form.

Given a form $\varphi(\tau, z) \in U_{t,b}$, for any $\hat{\kappa} \in \mathbb{Z}_{>0}$ we immediately obtain an element in $U_{\hat{\kappa}^2 t, \hat{\kappa} b}$ from the unwrapped wJf $\varphi(\tau, \hat{\kappa} z)$. (Note that this unwrapping is conceptually slightly different from the unwrapping described in section 2, since here we are unwrapping an object that is already a bona fide wJf.) For a given $U_{t,b}$, let us denote by $U_{t,b}^{\text{old}}$ the subspace generated by all such unwrapped forms coming from smaller index wJf.

The question we are really interested in is: For which values of $t$ and $b$ do genuinely new slow growing wJf appear that are not just unwrappings of lower index forms? And can these new slow growing wJf be described by unwrapped elliptic genera?

We therefore want to investigate the quotient space of new forms

$$U_{t,b}^{\text{new}} := U_{t,b}/U_{t,b}^{\text{old}}, \tag{4.4}$$

or equivalently, $U_{t,b} = U_{t,b}^{\text{new}} \oplus U_{t,b}^{\text{old}}$. Let us now give the precise form of our conjecture:

> **Conjecture:** For any $t, b$ such that $t/b \in \mathbb{Z}$ and $6b^2/t < 3$, there is a basis of $U_{t,b}^{\text{new}}$ which consists only of unwrapped elliptic genera of $\mathcal{N} = 2$ minimal models.

To put it another way: for any $U_{t,b}$ satisfying the conditions on $t, b$, we can find a basis consisting of 1) elements of $U_{t,b}^{\text{old}}$, that is unwrapped wJf, and 2) unwrapped elliptic genera of minimal models.

Note that this does *not* imply that any form in $U_{t,b}$ with $t, b$ satisfying (4.2) can be written as a linear combination of unwrapped elliptic genera: even though some of the unwrapped wJf in $U_{t,b}^{\text{old}}$ may indeed be unwrapped elliptic genera themselves, some of them may not. For instance, $\dim U_{36,4} = 3$, but its basis necessarily involves the unwrapped form $\varphi^{t=9,b=2}(\tau, 2z)$, which clearly cannot come from an elliptic genus, as $9/2 \notin \mathbb{Z}$.

We have tested this conjecture experimentally. In Table 2 we list all vector spaces $U_{t,b}$ with $t/b \in \mathbb{Z}$ and $6b^2/t < 3$ for $t \leq 18$ with a basis consisting of unwrapped elliptic genera. Moreover we have checked this conjecture up to $t = 50$, and found that it always holds. We thank Jason Quinones for providing us the data for high values of $t$ [57]. It would be interesting to prove the conjecture analytically and we hope to return to this question in the future.

Physically, the conjecture implies the following. Since the conditions we wrote down were necessary but not sufficient to be the elliptic genus of a physical 2d CFT, it was possible that there existed some weak Jacobi forms that was a mathematical "accident" and did not correspond to an actual physical spectrum (see e.g. [61, 62] for similar examples of this at the level

Table 2: $U_{t,b} \neq 0$ for $t \leq 18$ satisfying $t/b \in \mathbb{Z}$ and $6b^2/t < 3$. We provide a basis given by (unwrapped) elliptic genera of the minimal models defined in (3.8).

| $t$ | $b$ | $\frac{6b^2}{t}$ | dim $U_{t,b}$ | Basis |
|---|---|---|---|---|
| 3 | 1 | 2 | 1 | $\varphi^{D_4}(\tau, z)$ |
| 4 | 1 | $\frac{3}{2}$ | 1 | $\varphi^{A_3}(\tau, z)$ |
| 6 | 1 | 1 | 1 | $\varphi^{A_2}(\tau, z)$ |
| 10 | 2 | $\frac{12}{5}$ | 1 | $\varphi^{D_6}(\tau, z)$ |
| 12 | 2 | 2 | 2 | $\varphi^{A_5}(\tau, z), \varphi^{D_4}(\tau, 2z)$ |
| 16 | 2 | $\frac{3}{2}$ | 1 | $\varphi^{A_3}(\tau, 2z)$ |

of the partition function). Our conjecture implies that for $c < 3$, this does not happen, and the weak Jacobi forms that satisfy the conditions we list come from the ADE classification of the $\mathcal{N} = 2$ minimal models.

We note that there is a natural stronger form of the conjecture: Namely, that *all* $U_{t,b}^{\mathrm{new}}$ with $t/b \in \mathbb{Z}$ have a basis of unwrapped elliptic genera of $\mathcal{N} = (2,2)$ CFTs. We note that since [23] showed that if $b > \sqrt{t}$, the symmetric product cannot grow slowly, for such cases $U_{t,b} = 0$. This means that the CFTs in the stronger conjecture have to have $c \leq 6$. Since the only unitary $\mathcal{N} = (2,2)$ CFTs with $c < 3$ are minimal models, our original conjecture would follow immediately from the stronger form. Table 3 gives some examples of basis elements with $c \geq 3$. However, at higher values of $t$ there are examples of wJf for which we have not yet found a corresponding CFT. Since there is no classification of unitary $\mathcal{N} = (2,2)$ CFTs with $3 \leq c \leq 6$, it remains open if the stronger conjecture is correct.

## 4.2 Kazama-Suzuki theories and tensor product theories

So far, for $c < 3$ we could do a complete analysis due to the fact that all unitary $\mathcal{N} = (2,2)$ CFTs of such central charge are known and given by minimal models. Our analysis suggests that the range $3 \leq c \leq 6$ is just as interesting. There is however no classification of such $\mathcal{N} = (2,2)$ CFTs, and it is reasonable to expect that there is a very large number of them. We therefore cannot treat them systematically. Instead let us briefly discuss two types of constructions: so-called Kazama-Suzuki theories [63, 64], and tensor products of minimal models.

Kazama-Suzuki theories are a two-parameter family of rational $\mathcal{N} = (2,2)$ SCFTs given by the following coset

$$\frac{SU(M+1)_k \times SO(2M)_1}{SU(M)_{k+1} \times U(1)_{M(M+1)(M+k+1)}}, \tag{4.5}$$

for positive integers $k, M$. This coset theory has central charge

$$c = \frac{3kM}{k+M+1}. \tag{4.6}$$

There is a level-rank duality relating $k \leftrightarrow M$. Just as with $\mathcal{N} = 2$ minimal models, we can then assemble them into various modular invariant partition functions. For simplicity in what follows we take the diagonal invariant, corresponding to the $A_{k+1}$ family. For $M = 1$, these theories are then equivalent to the $A_{k+1}$ minimal model. However for $M > 1$, they are a natural generalization of minimal models. If $c \leq 6$, they are therefore natural candidates to test for slow-growing symmetric product elliptic genera.

Table 3: All $U_{t,b} \neq 0$ for $t \leq 18$ satisfying $t/b \in \mathbb{Z}$. The last column lists some CFTs, not necessarily minimal models, whose unwrapped elliptic genus can serve as a basis vector of $U_{t,b} \neq 0$. The last column is not necessarily an exhaustive list; it is possible that there are other CFTs whose (unwrapped) elliptic genera will give a complete set of basis vectors.

| $t$ | $b$ | $c = \frac{6b^2}{t}$ | CFT Examples |
|---|---|---|---|
| 1 | 1 | 6 | K3 sigma model |
| 2 | 1 | 3 | $T^2/\mathbb{Z}_2$ (see e.g. [31]) |
| 3 | 1 | 2 | $D_4$ |
| 4 | 1 | $\frac{3}{2}$ | $A_3$ |
| 4 | 2 | 6 | $T^4/G$ (see [31, 32]) |
| 6 | 1 | 1 | $A_2$ |
| 6 | 2 | 4 | $(A_2)^4$ |
| 8 | 2 | 3 | $(A_3)^2$ |
| 9 | 3 | 6 | $(A_2)^6$ |
| 10 | 2 | $\frac{12}{5}$ | $D_6$ |
| 12 | 2 | 2 | $A_5$ |
| 12 | 3 | $\frac{9}{2}$ | $(A_3)^3$ |
| 15 | 3 | $\frac{18}{5}$ | $(A_4)^2$ |
| 16 | 4 | 6 | $(A_3)^4$ |
| 18 | 3 | 3 | $(A_2)^3, A_2 \otimes A_5$ |

The elliptic genus for the $A_{k+1}$ Kazama-Suzuki models is given by [56]

$$Z_{\text{EG}}^{M,k}(\tau,z) = \prod_{j=1}^{M} \frac{\theta_1\left(\tau, \frac{(k+j)z}{M+k+1}\right)}{\theta_1\left(\tau, \frac{jz}{M+k+1}\right)}. \tag{4.7}$$

Note that $Z_{\text{EG}}^{M,k}(\tau,z) = Z_{\text{EG}}^{k,M}(\tau,z)$. As before, the function (4.7) is in general not a weak Jacobi form since it does not have integer charges. To get integer $U(1)_R$ charges, we rescale by $M+k+1$ if at least one of $M, k$ is even; otherwise we rescale by $2(M+k+1)$:

$$\varphi^{M,k}(\tau,z) = \begin{cases} \prod_{j=1}^{M} \frac{\theta_1(\tau,(k+j)z)}{\theta_1(\tau,jz)}, & Mk \in 2\mathbb{Z}, \\ \prod_{j=1}^{M} \frac{\theta_1(\tau,2(k+j)z)}{\theta_1(\tau,2jz)}, & Mk \notin 2\mathbb{Z}. \end{cases} \tag{4.8}$$

These weak Jacobi forms have

$$\begin{aligned} t &= \frac{kM(M+k+1)}{2}, \quad b = \frac{Mk}{2}, \quad &\text{if } Mk \in 2\mathbb{Z}, \\ t &= 2kM(M+k+1), \quad b = Mk, \quad &\text{if } Mk \notin 2\mathbb{Z}. \end{aligned} \tag{4.9}$$

Without loss of generality, let us assume that $M \leq k$. The Kazama-Suzuki theories with $c \leq 6$ are:

$$\begin{aligned} M &= 1, \quad k \geq 1, \\ M &= 2, \quad k \geq 2, \end{aligned}$$

$$M = 3, \quad k = 3, 4, 5, 6, 7, 8,$$
$$M = 4, \quad k = 4, 5. \tag{4.10}$$

For $M = 1$ we already know their symmetric products give slow growth since they are equivalent to $\mathcal{N} = 2$ minimal models. For $M = 2$ we have explicitly checked the symmetric product of Kazama-Suzuki up to $k = 10$, and find that they all give slow growth in the elliptic genus. It is thus natural to conjecture that all $M = 2$ Kazama-Suzuki theories satisfy this property. For the remaining cases in (4.10), the following pairs $(M, k)$ give a slow-growing weak Jacobi form in the symmetric product: $(3, 4), (3, 6), (3, 8), (4, 4), (4, 5)$. Interestingly, $(3, 3), (3, 5)$, and $(3, 7)$ do not. Moreover, every Kazama-Suzuki theory in (4.10) *except* for $(M, k) = (3, 3), (3, 5)$, and $(3, 7)$ has at least one single-trace twisted sector marginal operator. Thus it seems that Kazama-Suzuki theories have slow-growing elliptic genera if and only if they have at least one single-trace twisted sector modulus.

Another class of $\mathcal{N} = (2, 2)$ SCFTs that can have $c \leq 6$ are tensor products of minimal models. In particular, the tensor product of any two $\mathcal{N} = 2$ minimal models has $c < 6$. We have explicitly checked the first twelve $A$-series minimal models tensored with themselves, i.e.

$$(A_k)^2, \qquad k = 2, 3, \dots, 13, \tag{4.11}$$

and every one of them does give rise to both a slow-growing weak Jacobi form in the symmetric product, and has at least one exactly marginal single-trace twisted-sector operator. We have also shown various other tensor products of minimal models give rise to slow-growing weak Jacobi forms (see Table 3). On the other hand, not every tensor product of $\mathcal{N} = 2$ minimal models with $c \leq 6$ gives rise to a slow-growing weak Jacobi form. As an explicit example, the theory $(A_2)^5$ has $c = 5$ and its unwrapped elliptic genus is a weak Jacobi form with $t = 30$, $b = 5$ which we check does not obey (2.12). Moreover, the theory $(A_2)^5$ has no twisted-sector marginal operators. It would be interesting to classify which tensor products of minimal models do and do not give a slow-growing weak Jacobi form in the symmetric product and have single-trace exactly marginal twisted-sector operators.

## 4.3 Open questions

**Finding the supergravity duals and their string theory origin**

Two immediate follow-up questions to this work are: can we find a supergravity background in AdS$_3$ whose KK modes reproduce the signed count of BPS states we predict, and does there exist a top-down construction in string theory or M-theory whose low-energy approximation is this supergravity solution? In the D1D5 system, the signed 6d $(2, 0)$ supergravity KK spectrum on AdS$_3 \times S^3$ was found to precisely match the elliptic genus of Sym$^N$(K3) [33, 34]. Can we construct a supergravity background to match symmetric products of minimal models? Moreover in the D1D5 system, the BPS spectrum itself had interesting properties. The *unsigned* count of BPS states coming from supergravity KK modes differs substantially from the (signed) elliptic genus, with different asymptotics [65]. The signed count grows as $e^{\sqrt{n}}$ whereas the unsigned count grows as $e^{n^{3/4}}$. Is there a similar set of quarter-BPS states that fail to cancel at the supergravity point in moduli space?

**Slow-growing symmetric orbifold genera with seed central charge $3 \leq c \leq 6$**

So far in this paper we have mainly focused on $\mathcal{N} = (2, 2)$ SCFTs with $c < 3$, since there the classification of the theories is complete. Remarkably, every theory (i.e. the minimal models) exhibited both slow growth in the symmetric orbifold elliptic genus and had a single-trace marginal operator. In Section 4.2 we started exploring these features for certain classes of

theories with $3 \leq c \leq 6$. The situation there is more subtle – for instance, we gave explicit examples of theories with $3 \leq c \leq 6$ that give Hagedorn growth in the symmetric product elliptic genus (e.g. certain Kazama-Suzuki theories and minimal model tensor products). It would be interesting to extend this analysis further and make progress in understanding which seed theories with $3 \leq c \leq 6$ do and do not satisfy these two criteria. It would also be interesting to find examples of theories that satisfy one but not the other; or to prove that such a situation is impossible. We emphasize that in every example we have checked so far, either both criteria or neither criteria is satisfied. Another extension is to consider instead orbifolds by subgroups of $S_N$ which are known as permutation orbifolds. Many subgroups lead to a good large $N$ limit [66, 67] and it would interesting to see if one can find examples that also lead to supergravity-like growth.

**Nature of the strong coupling regime**

In this paper, we have provided evidence for a new infinite class of two-dimensional CFTs whose gravitational duals are described by semi-classical supergravity. The two main pieces of evidence are the slow growth of the elliptic genus and the existence of exactly marginal operators that turn on a "gauge" coupling between the $N$ copies of the theory. It is worthwhile mentioning how our theories, at strong coupling, could end up not being dual to semi-classical supergravity.

The main concern one could have is that the large coupling limit does not give parametrically large anomalous dimensions to the non-protected operators (in particular the operators of spin $s > 2$, which need to be heavy for the gravity dual to be described by Einstein gravity [4]). The existence of a single-trace marginal operators guarantees that a coupling can be turned on, but it remains a logical possibility that the anomalous dimensions remain bounded as $\lambda \to \infty$.

Apart from a direct inspection of the partition function at the strongly coupled point, one could diagnose this fact from the behavior of out-of-time-ordered correlators which detect chaotic behavior. At the orbifold point, the theory is not chaotic [68], and we can confidently claim that turning on the coupling will turn on chaos. However, a CFT dual to Einstein gravity is not just chaotic, it is maximally chaotic [69] and it remains possible that our theories do not saturate the chaos bound. If the anomalous dimensions did end up asymptoting to constants, the theory might resemble the two-dimensional supersymmetric versions of the SYK model discussed in [70], which are chaotic but not maximally chaotic. Nevertheless, we would like to emphasize that if such a situation occurs for our theories, there would need to be an additional conspiracy that is responsible for the slow-growth of the elliptic genus, which we find unlikely.

# Acknowledgements

We thank L. Eberhardt, M. Gaberdiel, J. Gauntlett, S. Kachru, I. Klebanov, J. Penedones, G. Sarosi, A. Sever, A. Sfondrini, S. Shenker, E. Silverstein, Y. Wang, S. Zhiboedov, X. Zhou, and M. Zimet for interesting discussions. We thank J. Quinones for sharing his results with us. We thank L. Eberhardt, M. Gaberdiel, R. Gopakumar, and S. Kachru for helpful comments on the draft. The work of A.B. is supported in part by the NWO VENI grant 680-47-464 / 4114. The work of N.B. is supported in part by the Simons Foundation Grant No. 488653. The work of A.C. is supported by the Delta ITP consortium, a program of the Netherlands Organisation for Scientific Research (NWO) that is funded by the Dutch Ministry of Education, Culture and Science (OCW). The work of S.M.H. is supported by the National Science and Engineering Council of Canada, an FRQNT new university researchers start-up grant, and the Canada Research Chairs program. The work of C.A.K. is supported in part by the Simons

Foundation Grant No. 629215. A.B. and A.C. thank KITP and the program "Gravitational Holography" for its hospitality during the completion of this work. This research was supported in part by the National Science Foundation under Grant No. NSF PHY-1748958.

# A    More on symmetric products of minimal models

## A.1    ADE series of minimal models

Let us define $\bar{m} = k + 2$. The minimal models then have central charge

$$c = 3 - \frac{6}{\bar{m}}. \tag{A.1}$$

Irreducible representation are given by $\mathcal{H}_{r,s}^{\pm}$ [48,49]. Here $\epsilon \in \{-1, 0, 1, 2\}$ determines the spin structure: in the NS sector, $\epsilon = 0, 2$, and in the Ramond sector, $\epsilon = \pm$ fixes the fermion parity $(-1)^F$ of the highest weight state of the representation. The $U(1)_R$ charges and dimension of the highest weight states in the Ramond sector are given by

$$h_{r,s}^{\epsilon} = \frac{r^2 - s^2}{4(k+2)} + \frac{c}{24}, \qquad Q_s^{\epsilon} = \frac{s}{k+2} + \frac{\epsilon}{2}. \tag{A.2}$$

The labels run as

$$r \in \{1, \dots k+1\}, \qquad 0 \le |s - \epsilon| \le r - 1, \qquad r + s \equiv 0 \pmod{2}. \tag{A.3}$$

From (A.2) we see that the BPS representations, that is the Ramond ground states, are given by $r = |s|$. We define the characters in the Ramond sector with $(-1)^F$ inserted,

$$\chi_{r,s}^{\pm}(\tau, z) = \mathrm{Tr}_{\mathcal{H}_{r,s}^{\pm}} (-1)^F y^{J_0} q^{L_0 - c/24}. \tag{A.4}$$

By defining

$$\chi_{r,s}^{\pm} = \chi_{\bar{m}-2-r, \bar{m}+s}^{\pm} = \chi_{r, s+2\bar{m}\mathbb{Z}}^{\pm}, \tag{A.5}$$

we can take $s$ to run over $\mathbb{Z}/2\bar{m}\mathbb{Z}$.

To turn this minimal model data into physical theories, we need to combine the characters into modular invariant partition functions

$$Z_{\mathrm{RR}}(\tau, \bar{\tau}, z, \bar{z}) := \mathrm{Tr}_{\mathrm{RR}} (-1)^{F_L + F_R} y^{J_0} \bar{y}^{\bar{J}_0} q^{L_0 - c/24} \bar{q}^{\bar{L}_0 - \bar{c}/24}, \tag{A.6}$$

as combinations of the minimal model characters $\chi_{r,s}^{\pm}$. Since $\mathcal{N} = 2$ minimal models can be written as cosets of $\hat{su}(2)$ models, we can use the ADE classification of [50] to obtain modular invariants of $\mathcal{N} = 2$ minimal models. This leads to an ADE series of minimal models, whose partition functions are given by [51]

$$Z^{\Phi}(\tau, \bar{\tau}, z, \bar{z}) = \sum_{\substack{r, s, \epsilon \\ r', s', \epsilon'}} N_{r,r'}^{\Phi} L_{s,s'} S_{\epsilon,\epsilon'} \chi_{r,s}^{\epsilon}(\tau, z) \chi_{r',s'}^{\epsilon'}(\bar{\tau}, \bar{z}). \tag{A.7}$$

Here the invariant $S$ fixes the spin structure, and $L$ is an invariant of a $\Theta$ system. In general, there are many possibilities for these invariants [52]. We require however that the theory has spectral flow: that is, we want the NS sector of the theory to be mapped to the R sector under spectral flow and vice versa. In string theory this criterion is usually imposed to ensure spacetime supersymmetry. In our case we need it to ensure that we can obtain the NS elliptic genus, whose growth we measure, can be obtained from the Ramond elliptic genus. Under this

Table 4: The ADE matrices $\Omega$ of Cappelli–Itzykson–Zuber [50].

| $\Phi$ | Coxeter number of $\Phi$ | $\Omega^{\Phi}$ |
|---|---|---|
| $A_{\bar{m}-1}$ | $\bar{m} = 1, 2, 3, \ldots$ | $\Omega_{\bar{m}}(1)$ |
| $D_{\bar{m}/2+1}$ | $\bar{m} = 6, 8, 10, \ldots$ | $\Omega_{\bar{m}}(1) + \Omega_{\bar{m}}(\bar{m}/2)$ |
| $E_6$ | 12 | $\Omega_{12}(1) + \Omega_{12}(4) + \Omega_{12}(6)$ |
| $E_7$ | 18 | $\Omega_{18}(1) + \Omega_{18}(6) + \Omega_{18}(9)$ |
| $E_8$ | 30 | $\Omega_{30}(1) + \Omega_{30}(6) + \Omega_{30}(10) + \Omega_{30}(15)$ |

requirement, $L$ and $S$ need to be chosen as diagonal [54], which implies that there is exactly one invariant for every $N^{\Phi}$, so that the ADE classification carries over [53].

Because of this invariance under spectral flow, it is enough to concentrate on the Ramond sector, or more precisely on the (——) structure. We define

$$\tilde{\chi}_s^r(\tau, z) := \chi_{r,s}^+ - \chi_{r,s}^-. \tag{A.8}$$

Explicit expressions for $\tilde{\chi}_s^r(\tau, z)$ can be found in, for instance, [37, 51].[4] In total we get for the RR partition function with $(-1)^F$ inserted

$$Z_{\text{RR}}^{\Phi}(\tau, \bar{\tau}, z, \bar{z}) = \frac{1}{2} \sum_{0 < r, r' < \bar{m}} N_{r,r'}^{\Phi} \sum_{s \in \mathbb{Z}/2\bar{m}} \tilde{\chi}_s^r(\tau, z) \tilde{\chi}_s^{r'}(\bar{\tau}, \bar{z}). \tag{A.9}$$

Here $\Phi$ is a simply laced Dynkin diagram with Coxeter number $\bar{m}$. Possible multiplicity matrices $N_{r,r'}^{\Phi}$ are in one-to-one correspondence to such Dynkin diagrams. The Capelli-Itzykson-Zuber (CIZ) matrix $N_{r,r'}^{\Phi}$ can be obtained from

$$N_{r,r'}^{\Phi} = \Omega_{r,r'}^{\Phi} - \Omega_{r,-r'}^{\Phi}, \tag{A.10}$$

where we introduced the $2\bar{m} \times 2\bar{m}$ matrix $\Omega^{\Phi}$. To specify the matrix $\Omega^{\Phi}$, we introduce for each divisor $n$ of $\bar{m}$ the matrix

$$\Omega_{\bar{m}}(n)_{r,r'} = \begin{cases} 1 & \text{if } r + r' \equiv 0 \pmod{2n} \text{ and } r - r' \equiv 0 \pmod{2\bar{m}/n}, \\ 0 & \text{otherwise.} \end{cases} \tag{A.11}$$

Note that the matrices satisfy $\Omega_{\bar{m}}(n)_{r,r'} = \Omega_{\bar{m}}(n)_{2\bar{m}-r,2\bar{m}-r'}$. From the definition we also immediately see that it makes sense to take the indices $r, r' \in \mathbb{Z}/2\bar{m}\mathbb{Z}$, which we will often do in the following. The $\Omega^{\Phi}$ can then be specified in terms of these matrices as in Table 4 .

We are actually not interested in the full partition function (A.9). Instead, we will consider two specializations. First, we recover the elliptic genus of the $\Phi$-type minimal model by setting $\bar{z} = 0$. By the usual argument, only BPS states make a contribution, meaning that

$$\tilde{\chi}_s^r(\tau, 0) = \delta_{r,s} - \delta_{r,-s}. \tag{A.12}$$

Defining $\tilde{\chi}_s^r(\tau, z) := -\tilde{\chi}_s^{-r}(\tau, z)$ for $r < 0$ and then continuing $r$ periodically in $2\bar{m}\mathbb{Z}$, and using $\Omega_{r,r'}^{\Phi} = \Omega_{-r,-r'}^{\Phi}$ we can write the elliptic genus as

$$Z_{\text{EG}}^{\Phi}(\tau, z) = \frac{1}{2} \sum_{r, r' \in \mathbb{Z}/2\bar{m}} \Omega_{r,r'}^{\Phi} \tilde{\chi}_{r'}^r(\tau, z) = \frac{1}{2} \text{Tr}(\Omega^{\Phi} \cdot \tilde{\chi}). \tag{A.13}$$

---

[4] Note that our convention for the $\tilde{\chi}_s^r$ is has a shift of $r$ by 1 compared to [37, 51].

We note that it is possible to compute the elliptic genus of the $\Phi$-type minimal model by exploiting the description of $\mathcal{N} = 2$ superconformal minimal models as IR fixed points of $\mathcal{N} = (2,2)$ Landau-Ginzburg theories in 2 dimensions with superpotential $W^{\Phi}$. Superpotentials in $\mathcal{N} = (2,2)$ theories in 2d are famously protected by a nonrenormalization theorem; a list of the superpotentials relevant for the $\mathcal{N} = 2$ minimal models is given in Table 4. As described in [55], the elliptic genus is an invariant of the 2d SQFT under renormalization group flow, and thus can be computed via a "free-field" computation in the UV Landau-Ginzburg description of the theory. This leads to the an expression in terms of free bosons and fermions for the elliptic genus of the $A$-type minimal models [55], and was subsequently generalized to the $D$- and $E$-type theories in [56], leading to the expression given in section 3.2.

Second, we recover the 1/2-BPS partition function $Z^{\Phi}_{\frac{1}{2}-\text{BPS}}$ by specializing to $q = 0, \bar{q} = 0$. Again only the BPS characters give a contribution, namely

$$
\tilde{\chi}^r_s(\tau, z)|_{q \to 0} = \begin{cases} y^{-\frac{1}{2}+\frac{r}{\bar{m}}} & s = r \pmod{2\bar{m}}, \\ -y^{\frac{1}{2}-\frac{r}{\bar{m}}} & s = -r \pmod{2\bar{m}}, \\ 0 & \text{otherwise.} \end{cases} \tag{A.14}
$$

giving

$$
Z^{\Phi}_{\frac{1}{2}-\text{BPS}} = \frac{1}{2} \sum_{0 < r < \bar{m}} N^{\Phi}_{r,r} \left( (y\bar{y})^{\frac{r}{\bar{m}}-\frac{1}{2}} + (y\bar{y})^{-\frac{r}{\bar{m}}+\frac{1}{2}} \right). \tag{A.15}
$$

If the CFT was a non-linear sigma model coming from a Calabi-Yau, this would be the Hodge diamond. We also note that the 1/2-BPS partition function only depends on the diagonal entries of the CIZ matrix and is always diagonal, even for the $D$ and $E$ series models.

## A.2 Moduli

It can be seen from (A.15) that the minimal models only have Ramond ground states with $Q = \bar{Q}$. This means that only $(c, c)$ and $(a, a)$ moduli appear. For concreteness we will focus on $(c, c)$, as by charge conjugation symmetry there is the same number of $(a, a)$ moduli. From (A.15), we see that Ramond ground states of charge

$$
Q_r = \frac{r}{k+2} - \frac{1}{2}, \qquad r = 1, \ldots, k+1, \tag{A.16}
$$

can appear in the theory. From (2.15) and (2.22), we see that to find chiral primaries of the right charge, we need to find configurations that satisfy

$$
Q = \sum_i \frac{2r_i - 2 + k(m_i - 1)}{2(k+2)} \overset{!}{=} 1, \tag{A.17}
$$

where $r_i$ is the representation, and $m_i$ the twist of the $i^{\text{th}}$ single trace factor.

### A-series

Let us describe how we count moduli in somewhat more detail for the $A$-series. For $k$ large enough, we will be able to write down closed form expressions for all moduli. This is due to the fact that, as can be seen from (A.17), the charges of twisted factors ($m_i > 1$) scale like $O(1)$ for large $k$, whereas untwisted factors scale like $O(1/k)$. This means that for $k$ large enough, only a fixed, small number of twisted factors can appear. Enumerating untwisted factors on the other hand is quite straightforward by counting partitions of integers.

For the $A$-series, this means that there are simple expressions once $k > 4$. For $k \leq 4$, additional moduli can appear. These can easily be found by an explicit computation; namely we have

$k = 1$, 29 marginal operators: 1 untwisted, 6 twist-2, 10 twist-3, 6 twist-4, 4 twist-5, 1 twist-6, 1 twist-7. There is one single trace operator for twist-5 and 7 each.

$k = 2$, 29 marginal operators: 3 untwisted; 13 twist-2; 9 twist-3; 3 twist-4; 1 twist-5. There is one single trace operator for twist-3,4,5 each.

$k = 4$, 33 marginal operators: 9 untwisted; 18 twist-2; 5 twist-3; 1 twist-4. There is one single trace operator for twist-2,3,4 each.

For $k$ large, we immediately see that that (A.17) cannot be satisfied if any of the $m_i > 3$. There are thus no moduli of twist bigger than 3 at sufficiently large $k$. In what follows we will use the notation

$$(r \dots r), \tag{A.18}$$

to mean a twisted sector whose length is given by the number of appearances of $r$. The value of $r$ is related to the charge in that sector according to (A.16). For example, (111) would be the vacuum of a twist-3 sector, while (2) would be an excited state in the untwisted sector. A straightforward counting of the possibilities gives:

$k$ even:

    **untwisted:** We have

$$\sum (r-1) = k+2, \qquad r-1 \leq k, \tag{A.19}$$

    which leads to $P(k+2) - 2$ possible moduli, the $-2$ coming from the fact that the partitions $\{k+2\}$ and $\{k+1,1\}$ are not allowed.

    **twist-2:** $(k/2+3, k/2+3), (33)(11), (22)(22), (22)(11)(2), (11)(11)(3), (11)(11)(2)(2)$. In addition we have $(rr)(i_1)\dots(i_K)$ where for a given $K$ we must have

$$\sum_{n=1}^{K} (i_n - 1) = \frac{k}{2} + 3 - r,$$

    giving an additional $\sum_{r=1}^{k/2+2} P(r)$ moduli.

    **twist-3:** $(333), (222)(2), (111)(2)(2), (111)(3)$.

$k$ odd:

    **untwisted:** The same $P(k+2) - 2$ untwisted moduli as for $k$ even.

    **twist-2:** $(33)(11), (22)(22), (22)(11)(2), (11)(11)(3), (11)(11)(2)(2)$

    **twist-3:** $(333), (222)(2), (111)(2)(2), (111)(3)$

In particular, we find that there is always one single trace twist 3 modulus, and for $k$ even also a single trace twist 2 modulus.

### D-series

The analysis of the $D$-series is therefore analogous to the $A$-series with $k$ even. The difference is now that $r$ only runs over odd integers,

$$r = 1, 3, \dots k+1. \tag{A.20}$$

Moreover there is an additional ground state with $r = k/2+1$, which we will denote by a hat if its value coincides with a value in (A.20). For $k$ large enough, we can thus read off the marginal operators:

$k = 4\rho$ :

> **untwisted:** We have $P(k/2)-1$ moduli without the $k/2+1$ state. We then have $P(k/4)$ states of the form $(k/2+1)(i_1)\dots(i_K)$, plus the state $(k/2+1)(k/2+1)(3)$, for a total of $P(k/2)+P(k/4)$ untwisted moduli.
>
> **twist-2:** $(k/2+3, k/2+3)$, $(33)(11)$, $(11)(11)(3)$, $(33)(k/2+1)$, $(11)(3)(k/2+1)$, $(k/2+1, k/2+1)(3)$ and $(rr)(i_1)\dots(i_K)$ where $\sum_{n}^{K}(i_n-1) = \frac{k}{2}+3-r$, giving an additional $\sum_{r=1}^{k/4} P(r)$ moduli, for a total of $5 + \sum_{r=1}^{k/4} P(r)$ twist-2 moduli.
>
> **twist-3:** $(333)$, $(111)(3)$.

$k = 4\rho + 2$ :

> **untwisted:** We have $P(k/2)-1$ moduli without the $k/2+1$ state, plus the state $(k/2+1)(k/2+1)(3)$, for a total of $P(k/2)$ untwisted moduli.
>
> **twist-2:** $(33)(11)$, $(11)(11)(3)$, $(33)(k/2+1)$, $(11)(3)(k/2+1)$, $(k/2+1, k/2+1)(3)$. Because $k/2$ is odd, there are no moduli of the form $(rr)(i_1)\dots(i_K)$.
>
> **twist-3:** $(333)$, $(111)(3)$.

### *E*-series

For the E-series minimal models, we find

$$
\chi^{E_6}_{\frac{1}{2}-\mathrm{BPS}}(y,\bar{y}) = (y\bar{y})^{-\frac{5}{12}} + (y\bar{y})^{-\frac{1}{6}} + (y\bar{y})^{-\frac{1}{12}} + (y\bar{y})^{\frac{1}{12}} + (y\bar{y})^{\frac{1}{6}} + (y\bar{y})^{\frac{5}{12}},
$$
$$
\chi^{E_7}_{\frac{1}{2}-\mathrm{BPS}}(y,\bar{y}) = (y\bar{y})^{-\frac{4}{9}} + (y\bar{y})^{-\frac{2}{9}} + (y\bar{y})^{-\frac{1}{9}} + 1 + (y\bar{y})^{\frac{1}{9}} + (y\bar{y})^{\frac{2}{9}} + (y\bar{y})^{\frac{4}{9}},
$$
$$
\chi^{E_8}_{\frac{1}{2}-\mathrm{BPS}}(y,\bar{y}) = (y\bar{y})^{-\frac{7}{15}} + (y\bar{y})^{-\frac{4}{15}} + (y\bar{y})^{-\frac{2}{15}} + (y\bar{y})^{-\frac{1}{15}} +
$$
$$
+ (y\bar{y})^{\frac{1}{15}} + (y\bar{y})^{\frac{2}{15}} + (y\bar{y})^{\frac{4}{15}} + (y\bar{y})^{\frac{7}{15}}. \tag{A.21}
$$

The number of moduli can be computed directly. It turns out that all three models have one single trace modulus of twist-2 of the form $(k/2+3, k/2+3)$.

These results are summarized in Table 1.

# B $\mathcal{N} = 2$ minimal models: Proof of slow growth

In this appendix we will prove that all $\mathcal{N} = 2$ minimal models satisfy the slow growth condition. The proof consists on verifying that (2.12) is strictly *non-positive* for the terms in the seed wJf $\varphi(\tau, z)$. As detailed in [23], there are a few simplifications that ease this task considerably:

1. Rewriting (2.12) as

$$
\alpha = \max_{j=0,\dots,b-1} \left[ -t\left(\frac{j}{b} - \frac{l}{2t}\right)^2 - \frac{1}{4t}\left(4tn - l^2\right) \right], \tag{B.1}
$$

   it is clear that we only need to check polar terms, i.e. terms with $4tn - l^2 < 0$.

2. A necessary, while not sufficient condition, to have $\alpha \leq 0$ is

$$
b^2 \leq t. \tag{B.2}
$$

   It is simple to show by checking that when $b^2 > t$ the most polar term, $y^b q^0$, has $\alpha > 0$.

3. Combining the two above requirements, with the properties of $c(n, l)$ and the allowed ranges of $j$, one can also show that it is sufficient to restrict

$$0 < l \le t. \tag{B.3}$$

All $\mathcal{N} = 2$ minimal models already have $b^2 \le t$, hence in the following we will focus on showing that their polar states with $0 < l \le t$ meet the slow growth criteria. The spectrum of these theories is dictated by the ADE classification of minimal models, as reflected in (3.8), and we will go through these cases individually.

## B.1 A-series, $k$ even

For the A-series with even $k$, we have

$$\varphi^{A_{k+1}}(\tau, z) = \frac{\theta_1(\tau, (k+1)z)}{\theta_1(\tau, z)}, \qquad \text{with} \quad b = \frac{k}{2}, \quad t = \frac{k(k+2)}{2}. \tag{B.4}$$

Therefore for a given polar term $q^n y^{-\ell}$, we have

$$\alpha = \max_{j=0,\dots,b-1}\left[-n + \frac{2j(\ell - j(k+2))}{k}\right]. \tag{B.5}$$

Let us suppose we fix the $q$-power $n$. In order to maximize $\alpha$, we should maximize $\ell$. Therefore it suffices to check $\alpha$ for the largest $\ell$ at fixed $n$. Using the product form (3.7) for the denominator of the wJf in (B.4), taking the $p^{\text{th}}$ term from the numerator, for fixed $n$ the largest $\ell$ comes from expanding the factor $(1 - yq)^{-1}$ that appears in the denominator of $\varphi^{A_{k+1}}$. If we take the $i^{\text{th}}$ term of this expansion and the $p^{\text{th}}$ term in the sum form of the numerator of (B.4), we find

$$n = \frac{p(p+1)}{2} + i, \tag{B.6}$$

and

$$\ell = p(k+1) + \frac{k}{2} + i. \tag{B.7}$$

For these states (B.5) reduces to

$$\alpha = \max_{j=0,\dots,b-1}\left[\frac{(p-2j)}{2k}(2(k+2)j - k(p+1)) + i\left(\frac{j}{b} - 1\right)\right]. \tag{B.8}$$

Since $j < b$, increasing values of $i$ lowers $\alpha$. Therefore, to find the maximum $\alpha$, it suffices to check only the cases with $i = 0$. Thus the problem now reduces to, for a fixed even $k$, show that

$$(p - 2j)(2(k+2)j - k(p+1)) \le 0, \tag{B.9}$$

for $p, j$ non-negative integers, with $0 \le p, j \le \frac{k}{2}$ due to (B.3). Proving this inequality splits naturally in two cases.

**Case 1: $p \ge 2j$.** The inequality (B.9) reduces to showing that

$$2(k+2)j - k(p+1) \le 0, \tag{B.10}$$

which we can write as

$$2(k+2)j - k(p+1) \le (k+2)p - k(p+1) = 2p - k \le 0, \tag{B.11}$$

since $p \le \frac{k}{2}$.

**Case 2: $p < 2j$.** Since $p$ and $j$ are integers, this case implies $2j \ge p + 1$. We can estimate the second parenthesis in (B.9) by

$$2(k+2)j - k(p+1) \ge 2(p+1) > 0. \tag{B.12}$$

Therefore $(p - 2j)(2(k+2)j - k(p+1)) \le 0$ for $p < 2j$.

## B.2  *A*-series, *k* odd

For the *A*-series with odd $k$, we have

$$\varphi^{A_{k+1}}(\tau, z) = \frac{\theta_1(\tau, 2(k+1)z)}{\theta_1(\tau, 2z)}, \qquad \text{with} \quad b = k, \quad t = 2k(k+2). \tag{B.13}$$

The proof here follows the same strategy as for $k$ even: For fix $n$, identify the largest value of $\ell$ and check that (B.1) is strictly negative for $\varphi^{A_{k+1}}$.

Let $p$ and $i$ be as above, such that

$$n = \frac{p(p+1)}{2} + i, \tag{B.14}$$

and

$$\ell = 2p(k+1) + k + 2i. \tag{B.15}$$

Note that we can assume $i \leq p$: if $i > p$, then we instead take the term with $p+1$ and $i-p-1$, which has the same $n$, but smaller $\ell$, since the restricted range (B.3) implies that $p \leq k$.[5] Plugging these values in for $\alpha$ gives

$$\alpha = \max_{j=0,\dots,k-1} \left[ \frac{i}{k}(2j-k) - \frac{(2j-p)}{2k}(k(2j-p) + 4j - k) \right]. \tag{B.16}$$

In contrast to (B.8), here $i$ can either increase or decrease the value of $\alpha$ which will require more work in proving our claim. In the following we will consider four cases dictated by the sign of based on the sign of $(2j-p)$ and $(2j-k)$, and show that (B.16) is non-positive for $k$ a positive odd integer, with integers satisfying $0 \leq j, p \leq k$ and $0 \leq i \leq p$.

**Case 1:** $2j < p \leq k$**.** Since we have

$$2i(2j-k) \leq 0, \quad 2j-p < 0, \tag{B.17}$$

it is sufficient to show that $(2j-p)k + 4j - k \leq 0$. From integrality,

$$2j-p \leq -1, \tag{B.18}$$

which implies

$$(2j-p)k + 4j - k \leq 2(2j-k) < 0. \tag{B.19}$$

**Case 2:** $p < 2j \leq k$**.** Due to

$$2i(2j-k) \leq 0, \quad 2j-p > 0, \tag{B.20}$$

it is sufficient to show that $(2j-p)k + 4j - k \geq 0$. From integrality,

$$2j-p \geq 1, \tag{B.21}$$

which implies

$$(2j-p)k + 4j - k \geq 4j \geq 0. \tag{B.22}$$

---

[5]We also note that for $0 \leq i \leq p$ there are a few values of $i$ that lead to non-polar terms. Decomposing the range of $i$ to accommodate for only polar terms is an unnecessary complication. To keep the inequalities simple our proof includes these non-polar states, in addition to all of the relevant polar states. A similar issue also occurs in the D-series, and it will be ignored there too.

**Case 3:** $p \leq k < 2j$. Since $2i(2j - k) > 0$, it suffices to show the inequality for the largest value of $i$, i.e., $i = p$. Then $\alpha$ reduces to

$$\alpha = \max_{j=0,\ldots,k-1} \left[ \frac{p}{k}(2j - k) - \frac{2j}{k}(2j - p) - \frac{(2j - p)}{2}(2j - p - 1) \right]. \tag{B.23}$$

It is clear that

$$\frac{(2j - p)}{2}(2j - p - 1) \geq 0, \tag{B.24}$$

and simple to verify that

$$p(2j - k) < 2j(2j - p), \tag{B.25}$$

for the conditions in this case. Therefore $\alpha$ is non-positive.

**Case 4:** $2j = p \leq k$. The second term of (B.16) vanishes, and $\alpha$ is clearly non-positive.

### B.3   $D$-series, $k \equiv 0 \pmod{4}$

For the $D$-series with $k \equiv 0 \pmod 4$, we have

$$\varphi^{D_{k/2+2}}(\tau, z) = \frac{\theta_1\left(\tau, \frac{k}{2}z\right)\theta_1\left(\tau, \frac{(k+4)}{4}z\right)}{\theta_1\left(\tau, \frac{k}{4}z\right)\theta_1(\tau, z)}, \qquad \text{with} \quad b = \frac{k}{4}, \quad t = \frac{k(k+2)}{8}. \tag{B.26}$$

To prove that all such functions satisfy $\alpha \leq 0$, we again look at all most polar terms at fixed $q$-exponent. By inspecting the $\theta$-functions appearing in $\varphi^{D_{k/2+2}}$, a useful way to write a given power $n$ is

$$n = \frac{p^2}{8} + i + \begin{cases} 0, & p \equiv 0 \pmod 4, \\ -\frac{1}{8}, & p \equiv 1 \pmod 4, \\ \frac{1}{2}, & p \equiv 2 \pmod 4, \\ -\frac{1}{8}, & p \equiv 3 \pmod 4, \end{cases} \tag{B.27}$$

for $p$ a positive integer and $i$ an integer satisfying

$$0 \leq i \leq \begin{cases} \frac{p-4}{4}, & p \equiv 0 \pmod 4, \\ \frac{p-1}{4}, & p \equiv 1 \pmod 4, \\ \frac{p-6}{4}, & p \equiv 2 \pmod 4, \\ \frac{p-3}{4}, & p \equiv 3 \pmod 4. \end{cases} \tag{B.28}$$

Note that for even cases we have $p > 2$, while odd instances have $p \geq 1$. Given this parametrization of $n$, we believe the most polar term at a given $n$ has $\ell$ given by at most

$$\ell \leq \frac{p(k+1)}{4} + i + \begin{cases} 0, & p \equiv 0 \pmod 4, \\ -\frac{1}{4}, & p \equiv 1 \pmod 4, \\ \frac{1}{2}, & p \equiv 2 \pmod 4, \\ -\frac{3}{4}, & p \equiv 3 \pmod 4. \end{cases} \tag{B.29}$$

As in the case of the even minimal models for the $A$-series, setting $i$ to be any nonzero number only decreases $\alpha$. Thus it suffices to show the $\alpha \leq 0$ for $i = 0$. Thus, the only values of $(n, \ell)$ we will need to consider are:

$$n = \frac{p^2}{8} + \begin{cases} 0, & p \equiv 0 \pmod 4, \\ -\frac{1}{8}, & p \equiv 1 \pmod 4, \\ \frac{1}{2}, & p \equiv 2 \pmod 4, \\ -\frac{1}{8}, & p \equiv 3 \pmod 4, \end{cases} \qquad \ell = \frac{p(k+1)}{4} + \begin{cases} 0, & p \equiv 0 \pmod 4, \\ -\frac{1}{4}, & p \equiv 1 \pmod 4, \\ \frac{1}{2}, & p \equiv 2 \pmod 4, \\ -\frac{3}{4}, & p \equiv 3 \pmod 4, \end{cases} \tag{B.30}$$

with $p$ a positive integer. Demanding (B.3) for all four cases in (B.30) reduces to $p \leq \frac{k}{2}$.

To finally establish that $\alpha \leq 0$, the task is very similar to the A-series: we evaluate (B.1) for every case in (B.30), and by taking into account the range of $p$, we show the non-positivity of $\alpha$. These steps are straightforward, but rather tedious to present in detail here.

### B.4  *D*-series, $k \equiv 2$ (mod 4)

Here we have

$$\varphi^{D_{k/2+2}}(\tau,z) = \frac{\theta_1(\tau,kz)\,\theta_1\left(\tau,\frac{(k+4)}{2}z\right)}{\theta_1\left(\tau,\frac{k}{2}z\right)\theta_1(\tau,2z)}, \qquad \text{with} \quad b = \frac{k}{2}, \quad t = \frac{k(k+2)}{2}. \tag{B.31}$$

As before, we write a given power $n$ as

$$n = \frac{p^2}{8} + i + \begin{cases} 0, & p \equiv 0 \ (\text{mod } 4), \\ -\frac{1}{8}, & p \equiv 1 \ (\text{mod } 4), \\ \frac{1}{2}, & p \equiv 2 \ (\text{mod } 4), \\ -\frac{1}{8}, & p \equiv 3 \ (\text{mod } 4), \end{cases} \tag{B.32}$$

for $p$ a positive integer and $i$ an integer satisfying

$$0 \leq i \leq \begin{cases} \frac{p-4}{4}, & p \equiv 0 \ (\text{mod } 4), \\ \frac{p-1}{4}, & p \equiv 1 \ (\text{mod } 4), \\ \frac{p-6}{4}, & p \equiv 2 \ (\text{mod } 4), \\ \frac{p-3}{4}, & p \equiv 3 \ (\text{mod } 4). \end{cases} \tag{B.33}$$

Note that for even cases we have $p > 2$, while odd instances have $p \geq 1$. Given this parametrization of $n$, we have checked that for a given $n$ the maximal value of $\ell$ is at most

$$\ell \leq \frac{p(k+1)}{2} + 2i + \begin{cases} 0, & p \equiv 0 \ (\text{mod } 4), \\ -\frac{1}{2}, & p \equiv 1 \ (\text{mod } 4), \\ 1, & p \equiv 2 \ (\text{mod } 4), \\ -\frac{3}{2}, & p \equiv 3 \ (\text{mod } 4). \end{cases} \tag{B.34}$$

By demanding $\ell \leq t$, we get $p \leq k$ for even $p$ and $p \leq k+1$ for odd $p$. The subsequent steps in the proof of $\alpha$ are straightforward with the information provided here.

### B.5  *E*-series

An explicit check, using Mathematica, of the last three functions in (3.8) show that they obey $\alpha \leq 0$.

## C  $\chi_\infty^{\text{NS}}$ for $\mathcal{N} = 2$ minimal models

In this section, we write explicit expressions for the low-lying spectrum counted by the elliptic genus of symmetric products of minimal models. This function is denoted as $\chi_\infty^{\text{NS}}(\tau,z)$ in [23], which gave an explicit expression for this spectrum in terms of the coefficients of the "seed" elliptic genus, $c(n,\ell)$ (see also [21]). In particular, if we are computing the low-lying spectra of $\text{Sym}^N(X)$ where $X$ has unwrapped elliptic genus $\varphi^X(\tau,z)$,

$$\varphi^X(\tau,z) = \sum_{n,\ell} c(n,\ell) q^n y^\ell, \tag{C.1}$$

then the states $\chi_\infty^{\mathrm{NS}}(\tau, z)$ are given by

$$\chi_\infty^{\mathrm{NS}}(\tau, z) = \prod_{h,\ell} \frac{1}{(1 - q^h y^\ell)^{f_{\mathrm{NS}}(h,\ell)}}, \tag{C.2}$$

where:

$$f_{\mathrm{NS}}(h, \ell) = \tilde{f}\left(h - \frac{b\ell}{2t}, \ell\right),$$

$$\tilde{f}(n, \ell) = f(n, \ell) - c(0, \ell) - \delta_{n,0} \sum_{m>0} c(0, \ell + bm),$$

$$f(n, \ell) = \begin{cases} \displaystyle\sum_{\hat{m} \in b\mathbb{Z} - \ell} c(0, \hat{m}) & n = 0, \\ \displaystyle\sum_{\hat{m} \in b\mathbb{Z}} c(-n\hat{m}/b - n^2 t/b^2, \hat{m}) & \ell = -\frac{nt}{b}, n > 0, \\ 0 & \text{otherwise.} \end{cases} \tag{C.3}$$

If the theory has a large-radius holographic dual, the expression $\chi_\infty^{\mathrm{NS}}$ should be interpreted as counting some supergravity KK states in the theory; for example in the D1D5 system, it is counting the 6d KK modes in $\mathrm{AdS}_3 \times S^3$ [33]. In this section we write out these functions explicitly for all minimal models.

## C.1  *A*-series *k* even

The theory has

$$t = \frac{k(k+2)}{2}, \qquad b = \frac{k}{2}, \tag{C.4}$$

with

$$\varphi^{A_{k+1}}(\tau, z) = \frac{\theta_1(\tau, (k+1)z)}{\theta_1(\tau, z)}. \tag{C.5}$$

The nonzero values of $f(n, \ell)$ are given by

$$f(0, \ell) = \begin{cases} 3 & \ell \equiv 0 \ (\mathrm{mod}\ \frac{k}{2}), \\ 2 & \ell \not\equiv 0 \ (\mathrm{mod}\ \frac{k}{2}), \end{cases}$$

$$f\left(\frac{kn}{2}, -\frac{k(k+2)n}{2}\right) = 3, \quad n \in \mathbb{Z}^+,$$

$$f\left(\frac{k(n-\frac{1}{2})}{2}, -\frac{k(k+2)(n-\frac{1}{2})}{2}\right) = 1, \quad n \in \mathbb{Z}^+,\ k \equiv 0 \ (\mathrm{mod}\ 4). \tag{C.6}$$

This implies the nonzero $\tilde{f}(n, \ell)$ are given by

$$\tilde{f}(0, \ell) = \begin{cases} 1, & 1 \le \ell < \frac{k}{2}, \\ 2, & \ell = \frac{k}{2} \\ 2, & \ell > \frac{k}{2},\ \ell \not\equiv 0 \ (\mathrm{mod}\ \frac{k}{2}), \\ 3, & \ell > \frac{k}{2},\ \ell \equiv 0 \ (\mathrm{mod}\ \frac{k}{2}), \end{cases}$$

$$\tilde{f}(n, \ell) = -1, \quad n \in \mathbb{Z}^+, \ -\frac{k}{2} \le \ell \le \frac{k}{2},$$

$$\tilde{f}\left(\frac{kn}{2}, -\frac{k(k+2)n}{2}\right) = 3, \quad n \in \mathbb{Z}^+,$$

$$\tilde{f}\left(\frac{k(n-\frac{1}{2})}{2}, -\frac{k(k+2)(n-\frac{1}{2})}{2}\right) = 1, \quad n \in \mathbb{Z}^+,\ k \equiv 0 \ (\mathrm{mod}\ 4). \tag{C.7}$$

Finally this implies the nonzero $f_{NS}(h, \ell)$ are given by

$$f_{NS}\left(\frac{\ell}{2(k+2)}, \ell\right) = \begin{cases} 1, & 1 \leq \ell < \frac{k}{2}, \\ 2, & \ell = \frac{k}{2}, \\ 2, & \ell > \frac{k}{2}, \ell \not\equiv 0 \pmod{\frac{k}{2}}, \\ 3, & \ell > \frac{k}{2}, \ell \equiv 0 \pmod{\frac{k}{2}}, \end{cases}$$

$$f_{NS}\left(n + \frac{\ell}{2(k+2)}, \ell\right) = -1, \qquad n \in \mathbb{Z}^+, \quad -\frac{k}{2} \leq \ell \leq \frac{k}{2},$$

$$f_{NS}\left(\frac{kn}{4}, -\frac{k(k+2)n}{2}\right) = 3, \qquad n \in \mathbb{Z}^+,$$

$$f_{NS}\left(\frac{k(n-\frac{1}{2})}{4}, -\frac{k(k+2)(n-\frac{1}{2})}{2}\right) = 1, \qquad n \in \mathbb{Z}^+, \quad k \equiv 0 \pmod 4. \tag{C.8}$$

Note that the last lines of (C.6), (C.7), and (C.8) are only if $k \equiv 0 \pmod 4$. We therefore get:

1. If $k \equiv 0 \pmod 4$:

$$\chi_\infty^{NS, A_{k+1}} = \left[\prod_{n=1}^\infty \frac{(1-q^n)}{(1-q^{\frac{n}{2(k+2)}} y^n)^2 (1-q^{\frac{kn}{4(k+2)}} y^{\frac{kn}{2}})(1-q^{\frac{kn}{4}} y^{-\frac{k(k+2)n}{2}})^3 (1-q^{\frac{k(n-\frac{1}{2})}{4}} y^{-\frac{k(k+2)(n-\frac{1}{2})}{2}})}\right] \times$$

$$\left[\prod_{n=1}^\infty \prod_{\ell=1}^{\frac{k}{2}} (1-q^{n-1+\frac{\ell}{2(k+2)}} y^\ell)(1-q^{n-\frac{\ell}{2(k+2)}} y^{-\ell})\right]. \tag{C.9}$$

2. If $k \equiv 2 \pmod 4$:

$$\chi_\infty^{NS, A_{k+1}} = \left[\prod_{n=1}^\infty \frac{(1-q^n)}{(1-q^{\frac{n}{2(k+2)}} y^n)^2 (1-q^{\frac{kn}{4(k+2)}} y^{\frac{kn}{2}})(1-q^{\frac{kn}{4}} y^{-\frac{k(k+2)n}{2}})^3}\right] \times$$

$$\left[\prod_{n=1}^\infty \prod_{\ell=1}^{\frac{k}{2}} (1-q^{n-1+\frac{\ell}{2(k+2)}} y^\ell)(1-q^{n-\frac{\ell}{2(k+2)}} y^{-\ell})\right]. \tag{C.10}$$

## C.2 A-series $k$ odd

The theory has

$$t = 2k(k+2), \qquad b = k, \tag{C.11}$$

with

$$\varphi^{A_{k+1}}(\tau, z) = \frac{\theta_1(\tau, 2(k+1)z)}{\theta_1(\tau, 2z)}. \tag{C.12}$$

The nonzero values of $f(n, \ell)$ are given by

$$f(0, \ell) = \begin{cases} 2, & \ell \equiv 0 \pmod k, \\ 1, & \ell \not\equiv 0 \pmod k, \end{cases}$$

$$f(kn, -2k(k+2)n) = 2, \qquad n \in \mathbb{Z}^+. \tag{C.13}$$

This implies the nonzero $\tilde{f}(n, \ell)$ are

$$\tilde{f}(0, \ell) = \begin{cases} 1, & \ell = k, \\ 1, & \ell > k, \ell \not\equiv 0 \pmod k, \\ 2, & \ell > k, \ell \equiv 0 \pmod k, \end{cases}$$

$$\tilde{f}(n,\ell) = -1, \qquad n \in \mathbb{Z}^+, \ -k \leq \ell \leq k, \ \ell \text{ odd},$$
$$\tilde{f}(kn, -2k(k+2)n) = 2, \qquad n \in \mathbb{Z}^+. \tag{C.14}$$

Finally this implies the nonzero $f_{\mathrm{NS}}(h, \ell)$ are given by

$$f_{\mathrm{NS}}\left(\frac{\ell}{4(k+2)}, \ell\right) = \begin{cases} 1, & \ell = k, \\ 1, & \ell > k, \ \ell \not\equiv 0 \ (\mathrm{mod} \ k), \\ 2, & \ell > k, \ \ell \equiv 0 \ (\mathrm{mod} \ k), \end{cases}$$

$$f_{\mathrm{NS}}\left(n + \frac{\ell}{4(k+2)}, \ell\right) = -1, \qquad n \in \mathbb{Z}^+, \ -k \leq \ell \leq k, \ \ell \text{ odd},$$

$$f_{\mathrm{NS}}\left(\frac{kn}{2}, -2k(k+2)n\right) = 2, \qquad n \in \mathbb{Z}^+. \tag{C.15}$$

We therefore get

$$\chi_\infty^{\mathrm{NS}, A_{k+1}} = \left[\prod_{\ell=k}^{\infty} \frac{1}{(1 - q^{\frac{\ell}{4(k+2)}} y^\ell)}\right] \left[\prod_{n=1}^{\infty} \frac{1}{(1 - q^{\frac{k(n+1)}{4(k+2)}} y^{k(n+1)})}\right] \times$$

$$\left[\prod_{n=1}^{\infty} \prod_{\ell=-\frac{(k-1)}{2}}^{\frac{k+1}{2}} (1 - q^{n + \frac{2\ell-1}{4(k+2)}} y^{2\ell-1})\right] \left[\prod_{n=1}^{\infty} \frac{1}{(1 - q^{\frac{kn}{2}} y^{-2k(k+2)n})^2}\right]. \tag{C.16}$$

### C.3 $D$-series $k \equiv 0 \ (\mathrm{mod} \ 4)$

The theory has

$$t = \frac{k(k+2)}{8}, \qquad b = \frac{k}{4}, \tag{C.17}$$

with

$$\varphi^{D_{k/2+2}}(\tau, z) = \frac{\theta_1\left(\tau, \frac{k}{2}z\right) \theta_1\left(\tau, \frac{(k+4)}{4}z\right)}{\theta_1\left(\tau, \frac{k}{4}z\right) \theta_1(\tau, z)}. \tag{C.18}$$

The nonzero values of $f(n, \ell)$ are given by

$$f(0, \ell) = \begin{cases} 4, & \ell \equiv 0 \ (\mathrm{mod} \ \frac{k}{4}), \\ 2, & \ell \not\equiv 0 \ (\mathrm{mod} \ \frac{k}{4}), \end{cases}$$

$$f\left(\frac{kn}{4}, -\frac{k(k+2)n}{8}\right) = 4, \qquad n \in \mathbb{Z}^+. \tag{C.19}$$

This implies the nonzero $\tilde{f}(n, \ell)$ are

$$\tilde{f}(0, \ell) = \begin{cases} 1, & 1 \leq \ell < \frac{k}{4}, \\ 3, & \ell = \frac{k}{4}, \\ 2, & \ell > \frac{k}{4}, \ \ell \not\equiv 0 \ (\mathrm{mod} \ \frac{k}{4}), \\ 4, & \ell > \frac{k}{4}, \ \ell \equiv 0 \ (\mathrm{mod} \ \frac{k}{4}), \end{cases}$$

$$\tilde{f}(n, 0) = -2, \qquad n \in \mathbb{Z}^+,$$

$$\tilde{f}(n, \ell) = -1, \qquad n \in \mathbb{Z}^+, \ -\frac{k}{4} \leq \ell \leq \frac{k}{4}, \ \ell \neq 0,$$

$$\tilde{f}\left(\frac{kn}{4}, -\frac{k(k+2)n}{8}\right) = 4, \qquad n \in \mathbb{Z}^+. \tag{C.20}$$

Finally this implies the nonzero $f_{NS}(h, \ell)$ are given by

$$f_{NS}(\frac{\ell}{k+2}, \ell) = \begin{cases} 1, & 1 \le \ell < \frac{k}{4}, \\ 3, & \ell = \frac{k}{4}, \\ 2, & \ell > \frac{k}{4}, \ell \not\equiv 0 \pmod{\frac{k}{4}}, \\ 4, & \ell > \frac{k}{4}, \ell \equiv 0 \pmod{\frac{k}{4}}, \end{cases}$$

$$f_{NS}(n, 0) = -2, \qquad n \in \mathbb{Z}^+,$$

$$f_{NS}(n + \frac{\ell}{k+2}, \ell) = -1, \qquad n \in \mathbb{Z}^+, \ -\frac{k}{4} \le \ell \le \frac{k}{4}, \ell \ne 0,$$

$$f_{NS}\left(\frac{kn}{8}, -\frac{k(k+2)n}{8}\right) = 4, \qquad n \in \mathbb{Z}^+. \tag{C.21}$$

We therefore get

$$\chi_\infty^{NS, D_{k/2+2}} = \left[\prod_{n=1}^\infty \frac{(1-q^n)^2}{(1-q^{\frac{kn}{4(k+2)}} y^{\frac{kn}{4}})^2 (1-q^{\frac{n}{k+2}} y^n)^2 (1-q^{\frac{kn}{8}} y^{-\frac{k(k+2)n}{8}})^4}\right] \times$$

$$\left[\prod_{n=1}^\infty \prod_{\ell=1}^{\frac{k}{4}} (1-q^{n+\frac{\ell}{k+2}-1} y^\ell)(1-q^{n-\frac{\ell}{k+2}} y^{-\ell})\right]. \tag{C.22}$$

## C.4 $D$-series $k \equiv 2 \pmod 4$

The theory has

$$t = \frac{k(k+2)}{2}, \qquad b = \frac{k}{2}, \tag{C.23}$$

with

$$\varphi^{D_{k/2+2}}(\tau, z) = \frac{\theta_1(\tau, kz) \theta_1\left(\tau, \frac{(k+4)z}{2}\right)}{\theta_1\left(\tau, \frac{kz}{2}\right) \theta_1(\tau, 2z)}. \tag{C.24}$$

The nonzero values of $f(n, \ell)$ are given by

$$f(0, \ell) = \begin{cases} 3, & \ell \equiv 0 \pmod{\frac{k}{2}} \\ 1, & \ell \not\equiv 0 \pmod{\frac{k}{2}} \end{cases}$$

$$f\left(\frac{kn}{2}, -\frac{k(k+2)n}{2}\right) = 3, \qquad n \in \mathbb{Z}^+. \tag{C.25}$$

This implies the nonzero $\tilde{f}(n, \ell)$ are given by

$$\tilde{f}(0, \ell) = \begin{cases} 2, & \ell = \frac{k}{2}, \\ 1, & \ell > \frac{k}{2}, \ell \not\equiv 0 \pmod{\frac{k}{2}}, \\ 3, & \ell > \frac{k}{2}, \ell \equiv 0 \pmod{\frac{k}{2}}, \end{cases}$$

$$\tilde{f}(n, \ell) = -1, \qquad n \in \mathbb{Z}^+, \ -\frac{k}{2} \le \ell \le \frac{k}{2}, \ell \text{ odd},$$

$$\tilde{f}(n, 0) = -1, \qquad n \in \mathbb{Z}^+,$$

$$\tilde{f}\left(\frac{kn}{2}, -\frac{k(k+2)n}{2}\right) = 3, \qquad n \in \mathbb{Z}^+. \tag{C.26}$$

Finally this implies the nonzero $f_{NS}(h,\ell)$ are given by

$$f_{NS}\left(\frac{\ell}{2(k+2)},\ell\right) = \begin{cases} 2, & \ell = \frac{k}{2}, \\ 1, & \ell > \frac{k}{2},\ \ell \not\equiv 0 \pmod{\frac{k}{2}}, \\ 3, & \ell > \frac{k}{2},\ \ell \equiv 0 \pmod{\frac{k}{2}}, \end{cases}$$

$$f_{NS}\left(n + \frac{\ell}{2(k+2)},\ell\right) = -1, \qquad n \in \mathbb{Z}^+,\ -\frac{k}{2} \le \ell \le \frac{k}{2},\ \ell \text{ odd},$$

$$f_{NS}(n,0) = -1, \qquad n \in \mathbb{Z}^+,$$

$$f_{NS}\left(\frac{kn}{4}, -\frac{k(k+2)n}{2}\right) = 3, \qquad n \in \mathbb{Z}^+. \tag{C.27}$$

We therefore get

$$\chi_\infty^{NS, D_{k/2+2}} = \left[\prod_{\ell=\frac{k}{2}+1}^{\infty} \frac{1}{(1 - q^{\frac{\ell}{2(k+2)}} y^\ell)}\right] \left[\prod_{n=1}^{\infty} \frac{1 - q^n}{(1 - q^{\frac{kn}{4(k+2)}} y^{\frac{kn}{2}})^2 (1 - q^{\frac{kn}{4}} y^{-\frac{k(k+2)n}{2}})^3}\right] \times$$

$$\left[\prod_{n=1}^{\infty} \prod_{\ell=-\frac{k-2}{4}}^{\frac{k+2}{4}} (1 - q^{n+\frac{2\ell-1}{2(k+2)}} y^{2\ell-1})\right]. \tag{C.28}$$

## C.5 $E_6$

The theory has

$$t = 60, \qquad b = 5, \tag{C.29}$$

with

$$\varphi^{E_6}(\tau, z) = \frac{\theta_1(\tau, 8z)\theta_1(\tau, 9z)}{\theta_1(\tau, 4z)\theta_1(\tau, 3z)}. \tag{C.30}$$

The nonzero values of $f(n,\ell)$ are given by

$$f(0,\ell) = \begin{cases} 2, & \ell \equiv 0 \pmod 5, \\ 1, & \ell \not\equiv 0 \pmod 5, \end{cases}$$

$$f(5n, -60n) = 2, \qquad n \in \mathbb{Z}^+. \tag{C.31}$$

This implies the nonzero $\tilde{f}(n,\ell)$ are given by

$$\tilde{f}(0,\ell) = \begin{cases} 1, & \ell = 5, \\ 1, & \ell \ge 3,\ \ell \not\equiv 0 \pmod 5, \\ 2, & \ell > 5,\ \ell \equiv 0 \pmod 5, \end{cases}$$

$$\tilde{f}(n,\pm 5) = \tilde{f}(n,\pm 2) = \tilde{f}(n,\pm 1) = -1, \qquad n \in \mathbb{Z}^+,$$

$$\tilde{f}(5n, -60n) = 2, \qquad n \in \mathbb{Z}^+. \tag{C.32}$$

Finally this implies the nonzero $f_{NS}(h,\ell)$ are given by

$$f_{NS}\left(\frac{\ell}{24},\ell\right) = \begin{cases} 1, & \ell = 5, \\ 1, & \ell \ge 3,\ \ell \not\equiv 0 \pmod 5, \\ 2, & \ell > 5,\ \ell \equiv 0 \pmod 5, \end{cases}$$

$$f_{NS}\left(n \pm \frac{5}{24}, \pm 5\right) = f_{NS}\left(n \pm \frac{1}{12}, \pm 2\right) = f_{NS}\left(n \pm \frac{1}{24}, \pm 1\right) = -1, \qquad n \in \mathbb{Z}^+,$$

$$f_{\text{NS}}\left(\frac{5n}{2}, -60n\right) = 2, \qquad n \in \mathbb{Z}^+. \tag{C.33}$$

Where therefore get

$$\chi_{\infty}^{\text{NS},E_6} = \left(1 - q^{\frac{1}{24}} y\right)\left(1 - q^{\frac{1}{12}} y^2\right)\left(1 - q^{\frac{5}{24}} y^5\right) \times \tag{C.34}$$

$$\prod_{n=1}^{\infty} \frac{(1 - q^{n+\frac{5}{24}} y^5)(1 - q^{n-\frac{5}{24}} y^{-5})(1 - q^{n+\frac{1}{12}} y^2)(1 - q^{n-\frac{1}{12}} y^{-2})(1 - q^{n+\frac{1}{24}} y)(1 - q^{n-\frac{1}{24}} y^{-1})}{(1 - q^{\frac{5n}{2}} y^{-60n})^2(1 - q^{\frac{n}{24}} y^n)(1 - q^{\frac{5n}{24}} y^{5n})}.$$

## C.6 $E_7$

The theory has

$$t = 36, \qquad b = 4, \tag{C.35}$$

with

$$\varphi^{E_7}(\tau, z) = \frac{\theta_1(\tau, 6z)\theta_1(\tau, 7z)}{\theta_1(\tau, 2z)\theta_1(\tau, 3z)}. \tag{C.36}$$

The nonzero values of $f(n, \ell)$ are given by

$$f(0, \ell) = \begin{cases} 3, & \ell \equiv 0 \pmod 4, \\ 2, & \ell \equiv 2 \pmod 4, \\ 1, & \ell \equiv 1, 3 \pmod 4, \end{cases}$$
$$f(4n, -36n) = 3, \qquad n \in \mathbb{Z}^+,$$
$$f(4n - 2, -36n + 18) = 1, \qquad \in \mathbb{Z}^+. \tag{C.37}$$

This implies the nonzero $\tilde{f}(n, \ell)$ are given by

$$\tilde{f}(0, \ell) = \begin{cases} 1, & \ell = 2, \\ 1, & \ell \geq 3, \ \ell \text{ odd}, \\ 2, & \ell = 4, \\ 2, & \ell > 4, \ \ell \equiv 2 \pmod 4, \\ 3, & \ell > 4, \ \ell \equiv 0 \pmod 4, \end{cases}$$
$$\tilde{f}(n, \pm 4) = \tilde{f}(n, \pm 2) = \tilde{f}(n, \pm 1) = \tilde{f}(n, 0) = -1, \qquad n \in \mathbb{Z}^+,$$
$$\tilde{f}(4n, -36n) = 3, \qquad n \in \mathbb{Z}^+,$$
$$\tilde{f}(4n - 2, -36n + 18) = 1, \qquad n \in \mathbb{Z}^+. \tag{C.38}$$

Finally this implies the nonzero $f_{\text{NS}}(h, \ell)$ are given by

$$f_{\text{NS}}\left(\frac{\ell}{18}, \ell\right) = \begin{cases} 1, & \ell = 2, \\ 1, & \ell \geq 3, \ \ell \text{ odd}, \\ 2, & \ell = 4, \\ 2, & \ell > 4, \ \ell \equiv 2 \pmod 4, \\ 3, & \ell > 4, \ \ell \equiv 0 \pmod 4, \end{cases}$$
$$f_{\text{NS}}\left(n \pm \frac{2}{9}, \pm 4\right) = f_{\text{NS}}\left(n \pm \frac{1}{9}, \pm 2\right) = f_{\text{NS}}\left(n \pm \frac{1}{18}, \pm 1\right) = f_{\text{NS}}(n, 0) = -1, \qquad n \in \mathbb{Z}^+,$$
$$f_{\text{NS}}(2n, -36n) = 3, \qquad n \in \mathbb{Z}^+$$
$$f_{\text{NS}}(2n - 1, -36n + 18) = 1, \qquad n \in \mathbb{Z}^+. \tag{C.39}$$

We therefore get

$$\chi_\infty^{\text{NS},E_7} = \left(1-q^{\frac{1}{18}}y\right)\left(1-q^{\frac{1}{9}}y^2\right)\left(1-q^{\frac{2}{9}}y^4\right)\times \tag{C.40}$$

$$\prod_{n=1}^{\infty}\frac{(1-q^{n+\frac{2}{9}}y^4)(1-q^{n-\frac{2}{9}}y^{-4})(1-q^{n+\frac{1}{9}}y^2)(1-q^{n-\frac{1}{9}}y^{-2})(1-q^{n+\frac{1}{18}}y)(1-q^{n-\frac{1}{18}}y^{-1})(1-q^n)}{(1-q^{2n}y^{-36n})^3(1-q^{2n-1}y^{-36n+18})(1-q^{\frac{n}{18}}y^n)(1-q^{\frac{n}{9}}y^{2n})(1-q^{\frac{2n}{9}}y^{4n})}.$$

## C.7 $E_8$

The theory has

$$t = 105, \qquad b = 7, \tag{C.41}$$

with

$$\varphi^{E_8}(\tau,z) = \frac{\theta_1(\tau,12z)\theta_1(\tau,10z)}{\theta_1(\tau,5z)\theta_1(\tau,3z)}. \tag{C.42}$$

The nonzero values of $f(n,\ell)$ are given by

$$f(0,\ell) = \begin{cases} 2, & \ell \equiv 0 \ (\text{mod } 7), \\ 1, & \ell \not\equiv 0 \ (\text{mod } 7), \end{cases}$$

$$f(7n,-105n) = 2, \quad n \in \mathbb{Z}^+. \tag{C.43}$$

This implies the nonzero $\tilde{f}(n,\ell)$ are given by

$$\tilde{f}(0,\ell) = \begin{cases} 1, & \ell = 3,5,6,7, \\ 1, & \ell > 7, \ \ell \not\equiv 0 \ (\text{mod } 7), \\ 2, & \ell > 7, \ \ell \equiv 0 \ (\text{mod } 7), \end{cases}$$

$$\tilde{f}(n,\pm 7) = \tilde{f}(n,\pm 4) = \tilde{f}(n,\pm 2) = \tilde{f}(n,\pm 1) = -1, \qquad n \in \mathbb{Z}^+,$$

$$\tilde{f}(7n,-105n) = 2, \qquad n \in \mathbb{Z}^+. \tag{C.44}$$

Finally this implies the nonzero $f_{\text{NS}}(h,\ell)$ are given by

$$f_{\text{NS}}\left(\frac{\ell}{30},\ell\right) = \begin{cases} 1, & \ell = 3,5,6,7, \\ 1, & \ell > 7, \ \ell \not\equiv 0 \ (\text{mod } 7), \\ 2, & \ell > 7, \ \ell \equiv 0 \ (\text{mod } 7), \end{cases}$$

$$f_{\text{NS}}(n\pm\frac{7}{30},\pm 7) = f_{\text{NS}}(n\pm\frac{2}{15},\pm 4) = f_{\text{NS}}(n\pm\frac{1}{15},\pm 2) = f_{\text{NS}}(n\pm\frac{1}{30},\pm 1) = -1, \quad n \in \mathbb{Z}^+,$$

$$f_{\text{NS}}\left(\frac{7n}{2},-105n\right) = 2, \qquad n \in \mathbb{Z}^+. \tag{C.45}$$

We therefore get

$$\chi_\infty^{\text{NS},E_8} = \left(1-q^{\frac{1}{30}}y\right)\left(1-q^{\frac{1}{15}}y^2\right)\left(1-q^{\frac{2}{15}}y^4\right)\left(1-q^{\frac{7}{30}}y^7\right)\times$$

$$\left[\prod_{n=1}^{\infty}\frac{(1-q^{n+\frac{7}{30}}y^7)(1-q^{n-\frac{7}{30}}y^{-7})(1-q^{n+\frac{2}{15}}y^4)(1-q^{n-\frac{2}{15}}y^{-4})}{(1-q^{\frac{7n}{2}}y^{-105n})^2(1-q^{\frac{n}{30}}y^n)(1-q^{\frac{7n}{30}}y^{7n})}\times\right.$$

$$\left.(1-q^{n+\frac{1}{15}}y^2)(1-q^{n-\frac{1}{15}}y^{-2})(1-q^{n+\frac{1}{30}}y)(1-q^{n-\frac{1}{30}}y^{-1})\right]. \tag{C.46}$$

# D   Large $N$ scaling of marginal operators

In this section, we review the relevant $N$-power counting for holographic CFTs, assuming that we have a planar-like limit where correlation functions factorize in the large $N$ limit. Note that a theory like $\mathcal{N} = 4$ SYM has $N^2$ degrees of freedom, while a symmetric orbifold has $N$ degrees of freedom. In what follows, we always use $N$ as the order of the symmetric orbifold so the expression will be slightly different than for $\mathcal{N} = 4$ SYM. In large $N$ theories, the light operators whose dimensions don't scale with $N$ are divided into two classes: single trace and multi-trace. We will use the following notation: a single trace operator will be denoted $O$ while a K-trace operator will be denoted $: O^K :$. All operators we consider have unit two-point function.[6] What separates single and multi trace operators is the scaling of connected correlation functions. A single-trace operator has connected correlation functions that scale as

$$\langle O_1....O_n \rangle_c \sim N^{\frac{2-n}{2}}. \tag{D.1}$$

In particular, this means that all OPE coefficients between single trace operators are $1/\sqrt{N}$ suppressed. These statements are also valid if we chose different types of single-trace operators. On the contrary, multi-trace operators have correlation functions than can scale as

$$\langle : O^{K_1} : .... : O^{K_n} : \rangle_c \sim N^0. \tag{D.2}$$

In particular, OPE coefficients of three multi-trace operators with themselves scale as $N^0$ [71]. We will now proceed in two steps. First, we will discuss how big we can make the sources for a $K$-trace marginal operator while still preserving a planar-like limit, and we will then discuss what the effect of such deformations are.

## D.1   Large $N$ scaling for deformation operators

We would now like to understand what large $N$-factorization implies for the deformation of a large $N$ theory by a marginal operator. We will give a short review of the discussion in [72] and generalize it for multi-trace operators. We have in mind a deformation of the theory by

$$\delta S = \lambda N^{\frac{\beta}{2}} \int d^2x : O^K : (x). \tag{D.3}$$

We would like to keep $\lambda \sim \mathcal{O}(1)$, and the question is how big can we make $\beta$ without spoiling the planar-like structure. We will now show that the answer to this question is quite simple, we have

$$\beta = 2 - K. \tag{D.4}$$

To see this, consider a connected $n$-point function of operators

$$\langle O_1...O_n \rangle_c \sim N^{\frac{2-n}{2}}. \tag{D.5}$$

Now let us deform this correlation function by the deformation (D.3). We can easily compute the correlator in the deformed theory in terms of correlation functions of the undeformed theory (we drop the $c$ notation but we always just consider connected correlation functions)

$$\langle O_1...O_n \rangle_\lambda = \langle O_1...O_n \rangle + \lambda N^\beta \int \langle O_1...O_n : O^K : \rangle + \mathcal{O}(\lambda^2). \tag{D.6}$$

---

[6]Note that this is *not* the usual normalization for operators like the current or the stress tensor.

We now need to study the second term. To leading order in the large $N$ expansion, the $K$ elements of $O^K$ will split into smaller correlation functions

$$\langle O_1...O_n : O^K : \rangle = \langle O_1...O_{n_1}O\rangle .... \langle O_1...O_{n_K}O\rangle \sim \prod_i N^{\frac{2-n_i-1}{2}} \sim N^{\frac{K-n}{2}}, \tag{D.7}$$

where we used $\sum_i n_i = n$. Demanding that this is the same order as the correlator we started with (namely $N^{\frac{2-n}{2}}$), we arrive at

$$\beta = 2 - K, \tag{D.8}$$

as advertised. Note that we have kept the leading possible piece where $O^K$ spreads into $K$ correlators. This contribution may yield zero in most of the correlators of the theory, but it will always give a non-zero answer in some correlators which is what fixes $\beta$.

## D.2 The effect of the deformations

We are now ready to study the effect of the deformations. Let us start by considering single-trace operators. Say we want to compute the deformation to order $\lambda^2$ of some 2-point function of $O_p$. We have

$$\langle OO\rangle_\lambda = \langle OO\rangle + \lambda N^{\frac{1}{2}} \int \langle O_p O_p O\rangle + \lambda^2 N \int \int \langle O_p O_p OO\rangle. \tag{D.9}$$

We can now see that the first two corrections produce $\mathcal{O}(1)$ contributions. The term linear in $\lambda$ has a power of $N^{\frac{1}{2}}$ but the the three-point function of single-trace operators scales as $N^{-\frac{1}{2}}$ so we obtain an order one scaling. Similarly, the second term receives two contributions:

$$\lambda^2 N \int \int \langle O_p O_p OO\rangle = \lambda^2 N \int \int \left( \langle O_p O_p\rangle \langle OO\rangle + \langle O_p O_p OO\rangle_c \right). \tag{D.10}$$

The first term can clearly not yield anomalous dimension since the 2 integrals will factor out and we will not get log terms. The second term can (and will) give log terms and hence an anomalous dimension to $O$. We see that the piece of interest will be $\mathcal{O}(1)$ since the connected correlator goes like $N^{-1}$ which will cancel against the $N$ out front.

Now consider a double trace deformation:

$$\langle O_p O_p\rangle_\lambda = \langle O_p O_p\rangle + \lambda \int \langle O_p O_p : O^2 : \rangle$$

$$= \langle O_p O_p\rangle + \lambda \int \langle O_p O\rangle \langle O_p O\rangle + \mathcal{O}(N^{-1}). \tag{D.11}$$

We can now see very clearly that the effect is very different. First if the probe $O_p \neq O$, then there is no $\mathcal{O}(N^0)$ and all anomalous dimensions are suppressed by $1/N$. As advocated in the main text, the effect of multi-trace deformations are small. Note however that we can produce order $N^0$ anomalous dimensions to $O$ itself from such a deformation. This is in accordance with the fact that double-trace deformations only change the boundary condition for the bulk fields, but that the rest of the theory remains unchanged. We conclude from this that double-trace deformations that preserve the 't Hooft limit cannot lift the spectrum, since their effect on other operators than $O$ is "quantum" and thus $1/N$ suppressed.

For higher-trace operators, the source itself is at least $1/\sqrt{N}$ suppressed and since all correlation functions are at most $\mathcal{O}(1)$, the anomalous dimension due to higher-trace deformations are suppressed. This type of deformation can still produce effects which change the leading behavior of connected correlators: In general, deforming by a $K$-trace operator can produce leading order effects on connected $K$-point function and higher (which corresponds to tree-level processes in the bulk), but these effects are purely quantum from the point of view of the spectrum.

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
