# Peer review of "$\mathcal{N}=2$ Minimal Models: A Holographic Needle in a Symmetric Orbifold Haystack"

_SciPost Physics, doi:SciPost Phys. 8, 084 (2020)_

## Round 2 · Referee Report · Anonymous (Referee 1) · 2020-4-29

Strengths
- The paper is well written
- It investigates an interesting physical problem, i.e. identifying interesting holographic setups in AdS/CFT
- The results are new and interesting
- The authors also study in detail the moduli of the relevant models
Weaknesses
- The authors stop short of trying to construct the holographic duals of the CFTs that they identify
Report
This is a very interesting and well written paper. The weaknesses that I highlighted are very minor. In particular, it would be extremely interesting to try and investigate what are the gravitational theories dual to the models studied here by the the authors but, even without this analysis, the article contains very interesting results and it is well worth publishing.
Requested changes
No changes are requested.

---

## Round 2 · Referee Report · Anonymous (Referee 2) · 2020-5-13

Strengths
- Novel and interesting results
- Well-written and clearly explained
- Background and motivation clearly stated
- Opens up interesting avenues of further reserch
Weaknesses
no weaknesses identified
Report
This paper explores the question of which types of CFTs can potentially have a supergravity dual within the framework of the AdS/CFT correspondence. The authors begin by pointing out that it is relatively easy to construct CFTs whose spectrum of states have a Hagedorn-type exponential growth, but this is not the expected behaviour of a supergravity theory. Rather, a supergravity theory would give rise to a slower growth, of the form $\rho(\Delta) \sim e^{c \Delta^{1/n}}$, where $n >1$ ($n=1$ for Hagedorn).
The authors resolve to search for candidate CFTs which have the desired growth properties. They restrict their attention to CFTs that are symmetric orbifolds $X^N/S_N$ of some seed CFT $X$. It turns out that if $X$ has central charge $c>6$ they will always exhibit Hagedorn growth and so fall outside of the realm of candidate theories. They further restrict their search to the regime where $c\leq 3$ for which a classification is known: these are the $\mathcal{N}=2$ minimal models. The regime $3\leq c\leq 6$ (for which no classification is known) is mainly left for future work. Some results are presented for Kazama-Suzuki models (which appear to exhibit slow growth except for some isolated cases) and tensor products of minimal models (not all of which exhibit slow growth).
The main result of the paper is surprising and somewhat remarkable: It turns out that the symmetric orbifolds of \emph{any} (unitary) $\mathcal{N}=2$ minimal models satisfies the required condition for being a candidate CFT with a supergravity dual. The authors show this essentially by analyzing the elliptic genera of these theories and showing that they have the desired growth properties. An implication of this result is thus a proposal for a new infinite family of potential holographic CFTs.
The authors also address the very relevant question: does every weak Jacobi form with a slow growth correspond to an elliptic genus of an $\mathcal{N}=2$ CFT? More precisely, one could at most expect that the elliptic genera of minimal models give a basis for such ``slow-growing'' Jacobi forms. They propose a precise conjecture concerning this and provide quite convincing numerical evidence. An analytic proof is left for future work.
The paper is well-written and provides a pleasant read. The results and methods are clearly explained and motivated. The main results are moreover extremely interesting and provide a crucial steps toward deeper explorations into the space of holographic CFTs.
For these reasons I recommend publication in its present form.
The authors resolve to search for candidate CFTs which have the desired growth properties. They restrict their attention to CFTs that are symmetric orbifolds $X^N/S_N$ of some seed CFT $X$. It turns out that if $X$ has central charge $c>6$ they will always exhibit Hagedorn growth and so fall outside of the realm of candidate theories. They further restrict their search to the regime where $c\leq 3$ for which a classification is known: these are the $\mathcal{N}=2$ minimal models. The regime $3\leq c\leq 6$ (for which no classification is known) is mainly left for future work. Some results are presented for Kazama-Suzuki models (which appear to exhibit slow growth except for some isolated cases) and tensor products of minimal models (not all of which exhibit slow growth).
The main result of the paper is surprising and somewhat remarkable: It turns out that the symmetric orbifolds of \emph{any} (unitary) $\mathcal{N}=2$ minimal models satisfies the required condition for being a candidate CFT with a supergravity dual. The authors show this essentially by analyzing the elliptic genera of these theories and showing that they have the desired growth properties. An implication of this result is thus a proposal for a new infinite family of potential holographic CFTs.
The authors also address the very relevant question: does every weak Jacobi form with a slow growth correspond to an elliptic genus of an $\mathcal{N}=2$ CFT? More precisely, one could at most expect that the elliptic genera of minimal models give a basis for such ``slow-growing'' Jacobi forms. They propose a precise conjecture concerning this and provide quite convincing numerical evidence. An analytic proof is left for future work.
The paper is well-written and provides a pleasant read. The results and methods are clearly explained and motivated. The main results are moreover extremely interesting and provide a crucial steps toward deeper explorations into the space of holographic CFTs.
For these reasons I recommend publication in its present form.
Requested changes
- a minor typo in the text between eqs 2.21 and 2.22: mutli $\to$ multi

---

## Round 2 · Referee Report · Anonymous (Referee 3) · 2020-5-14

Strengths
- Well written paper summarizing both the known constraints and having new, interesting, results.
- The introduction is specially clear and provides a useful perspective on the subject.
Weaknesses
- None of note
Report
The paper examines the necessary conditions for large central charge CFTs to admit classical gravity duals, in particular, having a low energy spectrum with sub-Hagedorn entropy. The idea is to consider a simple class of CFTs and use the symmetric orbifold construction to generate a theory with large central charge. Generically, such symmmetric orbifold CFTs do not however satisfy the criteria required.
In this work, the authors identify that in the family of $\mathcal{N}=2$ minimal models one can achieve the necessary requirements, which in itself is an interesting statement. They furthermore go on to show that these models also have moduli that would allow for deformation away from the orbifold point (which itself would not qualify, as it would have higher spin conserved currents).
While they are unable to argue in the present work what the dual gravity theories would themselves be, these examples are nevertheless interesting and provide a useful starting point for further investigations.
I recommend the paper be accepted for publication.
In this work, the authors identify that in the family of $\mathcal{N}=2$ minimal models one can achieve the necessary requirements, which in itself is an interesting statement. They furthermore go on to show that these models also have moduli that would allow for deformation away from the orbifold point (which itself would not qualify, as it would have higher spin conserved currents).
While they are unable to argue in the present work what the dual gravity theories would themselves be, these examples are nevertheless interesting and provide a useful starting point for further investigations.
I recommend the paper be accepted for publication.
Requested changes
I find the paper clear and through and would recommend publishing it in its present form.

---

## Editorial Decision

published